# Model-based RL with Optimistic Posterior Sampling: Structural Conditions and Sample Complexity

**Alekh Agarwal**
Google Research

**Tong Zhang**
Google Research and HKUST

## Abstract

We propose a general framework to design posterior sampling methods for model-based RL. We show that the proposed algorithms can be analyzed by reducing regret to Hellinger distance in conditional probability estimation. We further show that optimistic posterior sampling can control this Hellinger distance, when we measure model error via data likelihood. This technique allows us to design and analyze unified posterior sampling algorithms with state-of-the-art sample complexity guarantees for many model-based RL settings. We illustrate our general result in many special cases, demonstrating the versatility of our framework.

## 1 Introduction

In model-based RL, a learning agent interacts with its environment to build an increasingly more accurate estimate of the underlying dynamics and rewards, and uses such estimate to progressively improve its policy. This paradigm is attractive as it is amenable to sample-efficiency in both theory [Kearns and Singh, 2002, Brafman and Tennenholtz, 2002, Auer et al., 2008, Azar et al., 2017, Sun et al., 2019, Foster et al., 2021] and practice [Chua et al., 2018, Nagabandi et al., 2020, Schrittwieser et al., 2020]. The learned models also offer the possibility of use beyond individual tasks [Ha and Schmidhuber, 2018], effectively providing learned simulators. Given the importance of this paradigm, it is vital to understand how and when can sample-efficient model-based RL be achieved.

Two key questions any model-based RL agent has to address are: i) how to collect data from the environment, given the current learning state, and ii) how to define the quality of a model, given the currently acquired dataset. In simple theoretical settings such as tabular MDPs, the first question is typically addressed through optimism, typically via an uncertainty bonus, while likelihood of the data under a model is a typical answer to the second. While the empirical literature differs on its answer to the first question in problems with complex state spaces, it still largely adopts likelihood, or a VAE-style approximation [e.g. Chua et al., 2018, Ha and Schmidhuber, 2018, Sekar et al., 2020], as the measure of model quality. On the other hand, the theoretical literature for rich observation RL is much more varied in the loss used for model fitting, ranging from a direct squared error in parameters [Yang and Wang, 2020, Kakade et al., 2020] to more complicated divergence measures [Sun et al., 2019, Du et al., 2021, Foster et al., 2021]. Correspondingly, the literature also has a variety of structural conditions which enable model-based RL.

In this paper, we seek to unify the theory of model-based RL under a general theoretical and algorithmic framework. We address the aforementioned two questions by always using data likelihood as a model quality estimate, irrespective of the observation complexity, and use an optimistic posterior sampling approach for data collection. For this approach, we define a *Bellman error decoupling* framework which governs the sample complexity of finding a near-optimal policy. Our main result establishes that when the decoupling coefficient in a model-based RL setting is small, our *Model-based Optimistic Posterior Sampling* algorithm (MOPS) is sample-efficient.

A key conceptual simplification in MOPS is that the model quality is always measured using likelihood, unlike other general frameworks [Du et al., 2021, Foster et al., 2021] which need to modify their loss functions for different settings. Practically, posterior sampling is relatively amenable to tractable implementation via ensemble approximations [see e.g. Osband et al., 2016a, Lu and Van Roy, 2017, Chua et al., 2018, Nagabandi et al., 2020] or sampling methods such as stochastic gradient Langevin dynamics [Welling and Teh, 2011]. This is in contrast with the version-space based optimization methods used in most prior works. We further develop broad structural conditions on the underlying MDP and model class used, under which the decoupling coefficient admits good bounds. This includes many prominent examples, such as

- Finite action problems with a small witness rank [Sun et al., 2019]
- Linear MDPs with infinite actions
- Small witness rank and linearly embedded backups (**new**)
- $Q$-type witness rank problems (**new**)
- Kernelized Non-linear Regulators [Kakade et al., 2020]
- Linear mixture MDPs [Modi et al., 2020, Ayoub et al., 2020]

Remarkably, MOPS simultaneously addresses all the scenarios listed here and more, only requiring one change in the algorithm regarding the data collection policy induced as a function of the posterior. The analysis of model estimation itself, which is shared across all these problems, follows from a common online learning analysis of the convergence of our posterior. Taken together, our results provide a coherent and unified approach to model-based RL, and we believe that the conceptual simplicity of our approach will spur future development in obtaining more refined guarantees, more efficient algorithms, and including new examples under the umbrella of statistical tractability.

## 2 Related work

**Model-based RL.** There is a rich literature on model-based RL for obtaining strong sample complexity results, from the seminal early works [Kearns and Singh, 2002, Brafman and Tennenholtz, 2002, Auer et al., 2008] to the more recent minimax optimal guarantees [Azar et al., 2017, Zanette and Brunskill, 2019]. Beyond tabular problems, techniques have also been developed for rich observation spaces with function approximation in linear [Yang and Wang, 2020, Ayoub et al., 2020] and non-linear [Sun et al., 2019, Agarwal et al., 2020, Uehara et al., 2021, Du et al., 2021] settings. Of these, our work builds most directly on that of Sun et al. [2019] in terms of the structural properties used. However, the algorithmic techniques are notably different. Sun et al. [2019] repeatedly solve an optimization problem to find an optimistic model consistent with the prior data, while we use optimistic posterior sampling, which scales to large action spaces unlike their use of a uniform randomization over the actions. We also measure the consistency with prior data in terms of the likelihood of the observations under a model, as opposed to their integral probably metric losses. Du et al. [2021] effectively reuse the algorithm of Sun et al. [2019] in the model-based setting, so the same comparison applies. We note that recent model-based feature learning works [Agarwal et al., 2020, Uehara et al., 2021] do measure model fit using log-likelihood, and some of their technical analysis shares similarities with our proofs, though there are notable differences in the MLE versus posterior sampling approaches. Finally, we note that a parallel line of work has developed model-free approaches for function approximation settings [Jiang et al., 2017, Du et al., 2021, Jin et al., 2021], and while our structural complexity measure is always smaller than the Bellman rank of Jiang et al. [2017], our model-based realizability is often a stronger assumption than realizability of just the $Q^\star$ function. For instance, Jin et al. [2020], Du et al. [2019] and Misra et al. [2019] do not model the entire transition dynamics in linear and block MDP models.

**Posterior sampling in RL.** Posterior sampling methods for RL, motivated by Thompson sampling (TS) [Thompson, 1933], have been extensively developed and analyzed in terms of their expected regret under a Bayesian prior by many authors [see e.g. Osband et al., 2013, Russo et al., 2017, Osband et al., 2016b] and are often popular as they offer a simple implementation heuristic through approximation by bootstrapped ensembles [Osband et al., 2016a]. Worst-case analysis of TS in RL settings has also been done for both tabular [Russo, 2019, Agrawal and Jia, 2017] and linear [Zanette et al., 2020] settings. Our work is most closely related to the recent Feel-Good Thompson Sampling strategy proposed and analyzed in [Zhang, 2021], and its extensions [Zhang et al., 2021, Agarwal and Zhang, 2022]. Note that our model-based setting does not require the two timestep strategy to solve a minimax problem, as was required in the model-free work of Agarwal and Zhang [2022].

**Model-based control.** Model-based techniques are widely used in control, and many recent works analyze the Linear Quadratic Regulator [see e.g. Dean et al., 2020, Mania et al., 2019, Agarwal et al., 2019, Simchowitz and Foster, 2020], as well as some non-linear generalizations [Kakade et al., 2020, Mania et al., 2020, Mhammedi et al., 2020]. Our framework does capture many of these settings as we demonstrate in Section 6.

**Relationship with Foster et al. [2021].** This recent work studies a broad class of decision making problems including bandits and RL. For RL, they consider a model-based framing, and provide upper and lower bounds on the sample complexity in terms of a new parameter called the decision estimation coefficient (DEC). Structurally, DEC is closely related to the Hellinger decoupling concept introduced in this paper (see Definition 2). However, while Hellinger decoupling is only used in our analysis (and the distance is measured to the true model), the DEC analysis of Foster et al. [2021] measures distance to a plug-in estimator for the true model, and needs complicated algorithms to explicitly bound the DEC. In particular, it is not known if posterior sampling is admissible in their framework, which requires more careful control of a minimax objective. The Hellinger decoupling coefficient, in contrast, admits conceptually simpler optimistic posterior sampling techniques.

## 3    Setting and Preliminaries

We study RL in an episodic, finite horizon Contextual Markov Decision Process (MDP) that is parameterized as $(\mathcal{X}, \mathcal{A}, \mathcal{D}, R_\star, P_\star)$, where $\mathcal{X}$ is a state space, $\mathcal{A}$ is an action space, $\mathcal{D}$ is the distribution over the initial context, $R_\star$ is the expected reward function and $P_\star$ denotes the transition dynamics. An agent observes a context $x^1 \sim \mathcal{D}$ for some fixed distribution $\mathcal{D}$.[1] At each time step $h \in \{1, \ldots, H\}$, the agent observes the state $x^h$, chooses an action $a^h$, observes $r^h$ with $\mathbb{E}[r^h \mid x_h, a_h] = R_\star^h(x_h, a_h)$ and transitions to $x^{h+1} \sim P_\star^h(\cdot \mid x^h, a^h)$. We assume that $x^h$ for any $h > 1$ always includes the context $x^1$ to allow arbitrary dependence of the dynamics and rewards on $x^1$. Following prior works [e.g. Jiang et al., 2017, Sun et al., 2019], we assume that $r^h \in [0, 1]$ and $\sum_{h=1}^H r^h \in [0, 1]$ to capture sparse-reward settings [Jiang and Agarwal, 2018]. We make no assumption on the cardinality of the state and/or action spaces, allowing both to be potentially infinite. We use $\pi$ to denote the agent's decision policy, which maps from $\mathcal{X} \to \Delta(\mathcal{A})$, where $\Delta(\cdot)$ represents probability distributions over a set. The goal of learning is to discover an optimal policy $\pi_\star$, which is always deterministic and conveniently defined in terms of the $Q_\star$ function [see e.g. Puterman, 2014, Bertsekas and Tsitsiklis, 1996]

$$\pi_\star^h(x^h) = \underset{a \in \mathcal{A}}{\arg\max}\, Q_\star^h(x^h, a), \quad Q_\star^h(x^h, a^h) = \mathbb{E}[r^h + \max_{a' \in \mathcal{A}} Q_\star^{h+1}(x^{h+1}, a') \mid x^h, a^h], \quad (1)$$

where we define $Q_\star^{H+1}(x, a) = 0$ for all $x, a$. We also define $V_\star^h(x^h) = \max_a Q_\star^h(x^h, a)$.

In the model-based RL setting of this paper, the learner has access to a model class $\mathcal{M}$ consisting of tuples $(P_M, R_M)$,[2] denoting the transition dynamics and expected reward functions according to the model $M \in \mathcal{M}$. For any model $M$, we use $\pi_M^h$ and $V_M^h$ to denote the optimal policy and value function, respectively, at level $h$ in the model $M$. We assume that like $V_\star^h$, $V_M^h$ also satisfies the normalization assumption $V_M^h(x) \in [0, 1]$ for all $M \in \mathcal{M}$. We use $\mathbb{E}^M$ to denote expectations evaluated in the model $M$ and $\mathbb{E}$ to denote expectations in the true model.

We make two common assumptions on the class $\mathcal{M}$.

**Assumption 1** (Realizability) *$\exists M_\star \in \mathcal{M}$ such that $P_\star^h = P_{M_\star}^h$ and $R_\star^h = R_{M_\star}^h$ for all $h \in [H]$.*

The realizability assumption ensures that the model estimation is well-specified. We also assume access to a planning oracle for any fixed model $M \in \mathcal{M}$.

**Assumption 2** (Planning oracle) *There is a planning oracle which given a model $M$, returns its optimal policy $\pi_M = \{\pi^h\}_{h=1}^H$: $\pi_M = \arg\max_\pi \mathbb{E}^{M,\pi}\left[\sum_{h=1}^H R_M^h(x^h, \pi^h(x^h))\right]$, where $\mathbb{E}^{M,\pi}$ is the expectation over trajectories obtained by following the policy $\pi$ in the model $M$.*

Given $M \in \mathcal{M}$, we define model-based Bellman error as:

$$\mathcal{E}_B(M, x^h, a^h) = Q_M^h(x^h, a^h) - (P_\star^h[r^h + V_M^{h+1}])(x^h, a^h), \quad (2)$$

---

[1]We intentionally call $x^1$ a context and not an initial state of the MDP as we will soon make certain structural assumptions which depend on the context, but take expectation over the states.

[2]We use $P_M$ and $R_M$ to denote the sets $\{P_M^h\}_{h=1}^H$ and $\{R_M^h\}_{h=1}^H$ respectively.

**Algorithm 1** Model-based Optimistic Posterior Sampling (MOPS) for model-based RL
***

**Require:** Model class $\mathcal{M}$, prior $p_0 \in \Delta(\mathcal{M})$, policy generator $\pi_{\text{gen}}$, learning rates $\eta, \eta'$ and optimism coefficient $\gamma$.
1: Set $S_0 = \emptyset$.
2: **for** $t = 1, \ldots, T$ **do**
3:   Observe $x_t^1 \sim \mathcal{D}$ and draw $h_t \sim \{1, \ldots, H\}$ uniformly at random.
4:   Let $L_s^h(M) = -\eta(R_M^h(x_s^h, a_s^h) - r_s^h)^2 + \eta' \ln P_M^h(x_s^{h+1} \mid x_s^h, a_s^h)$.   ▷ Likelihood function
5:   Define $p_t(M) = p(M|S_{t-1}) \propto p_0(M) \exp(\sum_{s=1}^{t-1}(\gamma V_M(x_s^1) + L_s^{h_s}(M)))$ as the posterior.
                                                                      ▷ Optimistic posterior sampling update
6:   Let $\pi_t = \pi_{\text{gen}}(h_t, p_t)$                                              ▷ policy generation
7:   Play iteration $t$ using $\pi_t$ for $h = 1, \ldots, h_t$, and observe $\{(x_t^h, a_t^h, r_t^h, x_t^{h+1})_{h=1}^{h_t}\}$
8:   Update $S_t = S_{t-1} \cup \{x_t^h, a_t^h, r_t^h, x_t^{h+1}\}$ for $h = h_t$.
9: **end for**
10: **return** $(\pi_1, \ldots, \pi_T)$.
***

where for a function $f : \mathcal{X} \times \mathcal{A} \times [0,1] \times \mathcal{X}$, we define $(P_M f)(x, a) = \mathbb{E}^M[f(x, a, r, x')|x, a]$. This quantity plays a central role in our analysis due to the simulation lemma for model-based RL.

**Lemma 1** (Lemma 10 of [Sun et al., 2019]). *For any distribution $p \in \Delta(\mathcal{M})$ and $x^1$, we have $\mathbb{E}_{M \sim p}\left[V^\star(x^1) - V^{\pi_M}(x^1)\right] = \mathbb{E}_{M \sim p}\left[\sum_{h=1}^H \mathbb{E}_{x^h, a^h \sim \pi_M | x^1} \mathcal{E}_B(M, x^h, a^h) - \Delta V_M(x^1),\right]$, where $\Delta V_M(x^1) = V_M(x^1) - V^\star(x^1)$.*

The Bellman error can in turn be related to Hellinger error in conditional probability estimation via the decoupling coefficient in Definition 2, which captures the structural properties of the MDP.

We use typical measures of distance between probability distributions to capture the error in dynamcis, and for any two distributions $P$ and $Q$ over samples $z \in \mathcal{Z}$, we denote $\text{TV}(P, Q) = 1/2 \mathbb{E}_{z \sim P} |dQ(z)/dP(z) - 1|$, $\text{KL}(P||Q) = \mathbb{E}_{z \sim P} \ln dP(z)/dQ(z)$ and $D_H(P, Q)^2 = \mathbb{E}_{z \sim P}(\sqrt{dQ(z)/dP(z)} - 1)^2$. We use $\Delta(S)$ to denote the space of all probability distributions over a set $S$ (under a suitable $\sigma$-algebra) and $[H] = \{1, \ldots, H\}$.

**Effective dimensionality.** We often consider infinite-dimensional maps $\chi(z_1, z_2)$ over a pair of inputs $z_1 \in \mathcal{Z}_1$ and $z_2 \in \mathcal{Z}_2$. We define the effective dimensionality of such maps as follows.

**Definition 1** (Effective dimension) *Given any measure $p$ over $\mathcal{Z}_1 \times \mathcal{Z}_2$, and feature map $\chi$, define:*

$$\Sigma(p, z_1) = \mathbb{E}_{z_2 \sim p(\cdot|z_1)} \chi(z_1, z_2) \otimes \chi(z_1, z_2), \quad K(\lambda) = \sup_{p, z_1} \text{trace}((\Sigma(p, z_1) + \lambda I)^{-1} \Sigma(p, z_1)).$$

*For any $\epsilon > 0$, define the effective dimension of $\chi$ as: $d_{\text{eff}}(\chi, \epsilon) = \inf_{\lambda > 0} \left\{ K(\lambda) : \lambda K(\lambda) \leq \epsilon^2 \right\}$.*

If $\dim(\chi) = d$, $d_{\text{eff}}(\chi, 0) \leq d$, and $d_{\text{eff}}(\chi, \epsilon)$ can more generally be bounded in terms of spectral decay assumptions (see e.g. Proposition 2 in Agarwal and Zhang [2022]).

# 4   Model-based Optimistic Posterior Sampling

We now describe our algorithm MOPS for model-based RL in Algorithm 1. The algorithm defines an optimistic posterior over the model class and acts according to a policy generated from this posterior. Specifically, the algorithm requires a prior $p_0$ over the model class and uses an optimistic model-error measure to induce the posterior distribution. We now highlight some of the salient aspects of our algorithm design.

**Likelihood-based dynamics prediction.** At each round, MOPS computes a likelihood over the space of models as defined in Line 4. The likelihood of a model $M$ includes two terms. The first term measures the squared error of the expected reward function $R_M$ in predicting the previously observed rewards. The second term measures the loss of the dynamics in predicting the observed states $x_s^{h+1}$, given $x_s^h$ and $a_s^h$ each previous round $s$. For the dynamics, we use negative log-likelihood as the loss, and the reward and dynamics terms are weighted by respective learning rates $\eta$ and $\eta'$. Prior works of Sun et al. [2019] and Du et al. [2021] use an integral probability metric divergence, and require a more complicated Scheffé tournament algorithmically to handle model fitting under total

variation unlike our approach. The more recent work of Foster et al. [2021] needs to incorporate the Hellinger distance in their algorithm. In contrast, we directly learn a good model in the Hellinger distance (and hence total variation) as our analysis shows, by likelihood driven sampling.

**Optimistic posterior updates.** Prior works in tabular [Agrawal and Jia, 2017], linear [Zanette et al., 2020] and model-free [Zhang et al., 2021, Agarwal and Zhang, 2022] RL make optimistic modifications to the vanilla posterior to obtain worst-case guarantees, and we perform a similar modification in our algorithm. Concretely, for every model $M$, we add the predicted optimal value in the initial context at all previous rounds $s$ to the likelihood term, weighted by a parameter $\gamma$ in Line 5. Subsequently, we define the posterior using this optimistic likelihood. As learning progresses, the posterior concentrates on models which predict the history well, and whose optimal value function predicts a large average value on the context distribution. Consequently any model sampled from the posterior has an optimal policy that either attains a high value in the true MDP $M^\star$, or visits some parts of the state space not previously explored, as the model predicts the history reasonably well. Incorporating this fresh data into our likelihood further sharpens our posterior, and leads to the typical *exploit-or-learn* behavior that optimistic algorithms manifest in RL.

**Policy Generator.** Given a sampling distribution $p \in \Delta(\mathcal{M})$ (which is taken as the optimistic posterior in MOPS), and a time step $h$, we assume access to a policy generator $\pi_{\text{gen}}$ that takes parameters $h$ and $p$, and returns a policy $\pi_{\text{gen}}(h, p) : \mathcal{X} \to \mathcal{A}$ (line 6). MOPS executes this policy up to a random time $h_t$ (line 7), which we denote as $(x^{h_t}, a^{h_t})|x^1 \sim \pi_{\text{gen}}(h_t, p)$. The MDP then returns the tuple $(x^{h_t}, a^{h_t}, r^{h_t}, x^{h_t+1})$, which is used in our algorithm to update the posterior. The choice of the policy generator plays a crucial role in our sample complexity guarantees, and we shortly present a decoupling condition on the generator which is a vital component of our analysis.

For the examples considered in the paper (see Section 6), policy generators that lead to good regret bounds are given as follows.

- $Q$-type problems: $\pi_{\text{gen}}(h, p)$ follows a sample from the posterior, $\pi_{\text{gen}}(h, p) = \pi_M, M \sim p$.
- $V$-type problems with finite actions: $\pi_{\text{gen}}(h, p)$ generates a trajectory up to $x^h$ using a single sample from posterior $\pi_M$ with $M \sim p$ and then samples $a^h \sim \text{Unif}(\mathcal{A})$.[3]
- $V$-type problems with infinite actions: $\pi_{\text{gen}}(h, p)$ draws two *independent* samples $M, M' \sim p$ from the posterior. It generates a trajectory up to $x^h$ using $\pi_M$, and samples $a^h$ using $\pi_{M'}(\cdot|x^h)$.

It is worth mentioning that for Q-type problems, $\pi_{\text{gen}}(h, p)$ does not depend on $h$. Hence we can replace random choice of $h_t$ by executing length $H$ trajectories in Algorithm 1 and using all the samples in the loss. This modification of MOPS has a better regret bound in terms of $H$ dependency.

## 5 Main Result

We now present the main structural condition that we introduce in this paper, which is used to characterize the quality of the policy generator used in Algorithm 1. We will present several examples of concrete models which can be captured by this definition in Section 6. The assumption is inspired by prior decoupling conditions [Zhang, 2021, Agarwal and Zhang, 2022] used in the analysis of contextual bandits and some forms of model-free RL.

**Definition 2** (Hellinger Decoupling of Bellman Error) *Let a distribution $p \in \Delta(\mathcal{M})$ and a policy $\pi(x^h, a^h|x^1)$ be given. For any $\epsilon > 0$, $\alpha \in (0, 1]$ and $h \in [H]$, we define the Hellinger decoupling coefficient $\text{dc}^h(\epsilon, p, \pi, \alpha)$ of an MDP $M^\star$ as the smallest number $c^h \geq 0$ so that for all $x^1$:*

$$\mathbb{E}_{M \sim p} \mathbb{E}_{(x^h, a^h) \sim \pi_M(\cdot|x^1)} \mathcal{E}_B(M, x^h, a^h) \leq \left( c^h \underset{M \sim p}{\mathbb{E}} \underset{(x^h, a^h) \sim \pi(\cdot|x^1)}{\mathbb{E}} \ell^h(M, x^h, a^h) \right)^\alpha + \epsilon,$$

*where $\ell^h(M, x^h, a^h) = D_H(P_M(\cdot|x^h, a^h), P_\star(\cdot|x^h, a^h))^2 + (R_M(x^h, a^h) - R_\star(x^h, a^h))^2$.*

Intuitively, the distribution $p$ in Definition 2 plays the role of our estimate for $M^\star$, and we seek a low regret for the optimal policies of models $M \sim p$, which is closely related to the model-based Bellman error under samples $x^h, a^h$ are drawn from $\pi^M$ (the LHS of Lemma 1). The decoupling inequality relates the Bellman error to the estimation error of $p$ in terms of mean-squared error of

---

[3]This can be extended to a more general experimental design strategy as we show in Appendix E.1.

the rewards and a Hellinger distance to the true dynamics $P_\star$. However, it is crucial to measure this error under distribution of the data which is used under model-fitting. The policy $\pi = \pi_{\mathrm{gen}}(h, p)$ plays this role of the data distribution, and is typically chosen in a manner closely related to $p$ in our examples. The decoupling inequality bounds the regret of $p$ in terms of estimation error under $x^h, a^h \sim \pi$, for all $p$, and allows us to find a good distribution $p$ via online learning.

For stating our main result, we define a standard quantity for posterior sampling, measuring how well the prior distribution $p_0$ used in MOPS covers the optimal model $M_\star$.

**Definition 3** (Prior around true model) *Given $\alpha > 0$ and $p_0$ on $\mathcal{M}$, define*
$$\omega(\alpha, p_0) = \inf_{\epsilon > 0} \left[ \alpha \epsilon - \ln p_0(\mathcal{M}(\epsilon)) \right],$$

*where* $\mathcal{M}(\epsilon) = \left\{ M \in \mathcal{M} : \sup_{x^1} \sup_{h, x^h, a^h} \tilde{\ell}^h(M, x^h, a^h) \leq \epsilon^2 \right\}$, *with* $\tilde{\ell}^h(M, x^h, a^h) = $ $\mathrm{KL}(P_\star(\cdot | x^h, a^h) || P_M(\cdot | x^h, a^h)) + (R_M(x^h, a^h) - R_\star(x^h, a^h))^2$.

Definition 3 implicitly uses model realizability to ensure that $\mathcal{M}(\epsilon)$ is non-empty for any $\epsilon > 0$. However, we note that our bound based on $\omega(\alpha, p_0)$ can still be applied even if the model is misspecified, whence the optimization over $\epsilon$ naturally gets limited above the approximation error. Understanding the dependence of such approximation error in the final bounds and instantiating it with various models of misspecification/corruption is an interesting direction for future work.

Before stating our main result, we state a useful property of $\omega(\alpha, p_0)$, which illustrates its behavior. It also says that for concrete problems, we can replace the KL ball by the better behaved Hellinger ball and only pay an extra logarithmic penalty.

**Lemma 2.** *If $\mathcal{M}$ is finite, $p_0$ is uniform over $\mathcal{M}$, and $M^\star \in \mathcal{M}$, then $\omega(\alpha, p_0) \leq \ln |\mathcal{M}|$ for all $\alpha > 0$. More generally, suppose $P^\star \ll P_0$ with $\|dP^\star/dP_0\|_\infty \leq B$ for any reference probability measure $P_0$, and that $\mathcal{M}$ admits an $\ell_\infty$-covering under the metric $\ell^h$ (given in Definition 2) for all $h \in [H]$, of a size $N(\epsilon)$ at a radius $\epsilon$. Then for any $\epsilon \leq 2/3$ such that $B \geq \log(6B^2/\epsilon)$, there exists a prior $p_0^\epsilon$ such that $\omega(\alpha, p_0^\epsilon) \leq \alpha \epsilon + \log N(\epsilon/(6 \log(B/\nu)))$, where $\nu = \epsilon/(6 \log(6B^2/\epsilon))$.*

The prior $p_0^\epsilon$ adds a small perturbutation to the dynamics in $\mathcal{M}$ in order to ensure the boundedness of the KL divergence to $P^\star$. Detailed choice of $p_0^\epsilon$ and definition of $\zeta(B)$ are based on the result comparing KL divergence and Hellinger distance in Sason and Verdú [2016, Theorem 9], and we provide a proof of Lemma 2 in Appendix C.

We now state the main result of the paper.

**Theorem 1** (Sample complexity under decoupling) *Under Assumptions 1 and 2, suppose that there exists $\alpha > 0$ such that for all $p$, $\mathrm{dc}^h(\epsilon, p, \pi_{\mathrm{gen}}(h, p), \alpha) \leq \mathrm{dc}^h(\epsilon, \alpha)$. Define*
$$\mathrm{dc}(\epsilon, \alpha) = \left( \frac{1}{H} \sum_{h=1}^{H} \mathrm{dc}^h(\epsilon, \alpha)^{\alpha/(1-\alpha)} \right)^{(1-\alpha)/\alpha}.$$

*If we take $\eta = \eta' = 1/6$ and $\gamma \leq 0.5$, then the following bound holds for* MOPS*:*
$$\sum_{t=1}^{T} \mathbb{E} \left[ V_\star(x_t^1) - \mathbb{E}_{M \sim p_t} V_M(x_t^1) \right] \leq \frac{\omega(3HT, p_0)}{\gamma} + 2\gamma T + HT \left[ \epsilon + (1-\alpha)(20H\gamma\alpha)^{\frac{\alpha}{(1-\alpha)}} \mathrm{dc}(\epsilon, \alpha)^{\frac{\alpha}{(1-\alpha)}} \right].$$

To simplify the result, we consider finite model classes with $p_0$ as the uniform prior on $\mathcal{M}$. For $\alpha \leq 0.5$, by taking $\gamma = \min(0.5, \sqrt{\ln |\mathcal{M}|/T}, (\ln |\mathcal{M}|/T)^{1-\alpha} \mathrm{dc}(\epsilon, \alpha)^{-\alpha}/H)$, we obtain
$$\frac{1}{T} \sum_{t=1}^{T} \mathbb{E} \left[ V_\star(x_t^1) - \mathbb{E}_{M \sim p_t} V_M(x_t^1) \right] = O\left( \min \left( H \left( \frac{\mathrm{dc}(\epsilon, \alpha) \ln |\mathcal{M}|}{T} \right)^\alpha, \sqrt{\frac{\ln |\mathcal{M}|}{T}} \right) + \epsilon H \right). \quad (3)$$

We note that in Theorem 1, the decoupling coefficient fully characterizes the structural properties of the MDP. Once $\mathrm{dc}(\epsilon, \alpha)$ is estimated, Theorem 1 can be immediately applied. We will instantiate this general result with concrete examples in Section 6. Definition 2 appears related to the decision estimation coefficient (DEC) of Foster et al. [2021]. As expalined in Section 2, our definition is only needed in the analysis, and more suitable to posterior sampling based algorithmic design. The definition is also related to the Bilinear classes model of Du et al. [2021], since the bilinear structures can be turned into decoupling results as we will see in our examples. Compared to these earlier results, our definition is more amenable algorithmically.

**Proof of Theorem 1**

We now give a proof sketch for Theorem 1. As in prior works, we start from bounding the regret of any policy $\pi_M$ in terms of a Bellman error term and an optimism gap via Lemma 1. We note that in the definition of Bellman error in Lemma 1, model $M$ being evaluated is the same model that also generates the data, and this coupling cannot be handled directly using online learning. This is where the decoupling argument is used, which shows that the coupled Bellman error can be bounded by a decoupled loss. In the decoupled loss, the data is generated according to $\pi_{\text{gen}}(h_t, p_t)$, while the model being evaluated is drawn from $p_t$ independently of data generation. This intuition is captured in the following proposition, proved in Appendix A.

**Proposition 1** (Decoupling the regret) *Under conditions of Theorem 1, the regret of Algorithm 1 at any round $t$ can be bounded, for any $\mu > 0$ and $\epsilon > 0$, as*

$$\mathbb{E}\left[V_\star(x_t^1) - \mathbb{E}_{M \sim p_t} V_M(x_t^1)\right] \leq \mathbb{E}\mathbb{E}_{M \sim p_t}\left[\mu H \ell^{h_t}(M, x_t^{h_t}, a_t^{h_t}) - \Delta V_M(x_t^1)\right]$$
$$+ H\left[\epsilon + (1-\alpha)(\mu/\alpha)^{-\alpha/(1-\alpha)}\mathrm{dc}(\epsilon, \alpha)^{\alpha/(1-\alpha)}\right].$$

The proposition involves error terms involving the observed samples $(x_t^{h_t}, a_t^{h_t}, r_t^{h_t}, x_t^{h_t+1})$, which our algorithm controls via the posterior updates. Specifically, we expect the regret to be small whenever the posterior has a small average error of models $M \sim p_t$, relative to $M_\star$. This indeed happens as evidenced by our next result, which we prove in Appendix B.

**Proposition 2** (Convergence of online learning) *With $\eta = \eta' = 1/6$ and $\gamma \leq 0.5$, MOPS ensures:*

$$\sum_{t=1}^{T} \mathbb{E}\mathbb{E}_{M \sim p_t}\left[0.3\eta\gamma^{-1}\ell^{h_t}(M, x_t^{h_t}, a_t^{h_t}) - \Delta V_M(x_t^1)\right] \leq \gamma^{-1}\omega(3HT, p_0) + 2\gamma T.$$

Armed with Proposition 1 and Proposition 2, we are ready to prove the main theorem as follows.

*Proof of Theorem 1.* Combining Propositions 1 and 2 with $\mu H = 0.3\eta/\gamma$ gives the desired result. $\square$

# 6  MDP Structural Assumptions and Decoupling Coefficients Estimates

Since Definition 2 is fairly abstract, we now instantiate concrete models where the decoupling coefficient can be bounded in terms of standard problem complexity measures. We give examples of $V$-type and $Q$-type decouplings, a distinction highlighted in many recent works [e.g. Jin et al., 2021, Du et al., 2021]. The $V$-type setting captures more non-linear scenarios at the expense of slightly higher algorithmic complexity, while $Q$-type is more elegant for (nearly) linear settings.

## 6.1  $V$-type decoupling and witness rank

Sun et al. [2019] introduced the notion of witness rank to capture the tractability of model-based RL with general function approximation, building on the earlier Bellman rank work of Jiang et al. [2017] for model-free scenarios. For finite action problems, they give an algorithm whose sample complexity is controlled in terms of the witness rank, independent of the number of states, and show that the witness rank is always smaller than Bellman rank for model-free RL. The measure is based on a quantity called *witnessed model misfit* that captures the difference between two probability models in terms of the differences in expectations they induce over test functions chosen from some class. We next state a quantity closely related to witness rank.

**Assumption 3** (Generalized witness factorization) *Let $\mathcal{F} = \{f(x, a, r, x') = r + g(x, a, x') : g \in \mathcal{G}\}$, with $g(x, a, x') \in [0, 1]$, be given. Then there exist maps $\psi^h(M, x^1)$ and $u^h(M, x^1)$, and a constant $\kappa \in (0, 1]$, such that for any context $x^1$, level $h$ and models $M, M' \in \mathcal{M}$, we have*

$$\kappa \mathcal{E}_B(M, M', h, x^1) \leq \left|\langle \psi^h(M, x^1), u^h(M', x^1)\rangle\right|$$
$$\leq \sup_{f \in \mathcal{F}} \mathbb{E}_{x^h \sim \pi_M | x^1} \mathbb{E}_{a^h \sim \pi_{M'}(x^h)} \left|(P_{M'}^h f)(x^h, a^h) - (P_\star^h f)(x^h, a^h)\right|, \quad \text{(Bellman domination)}$$

where $\mathcal{E}_B(M, M', h, x^1) = \underset{x^h \sim \pi_M, a^h \sim \pi_{M'}|x^1}{\mathbb{E}} \left[ \mathcal{E}_B(M', x^h, a^h) \right]$. *We assume that* $\|u^h(M, x^1)\|_2 \leq B_1$ *for all $M$ and $x^1$.*

Sun et al. [2019] define a similar factorization, but allow arbitrary dependence of $f$ on the reward to learn the full distribution of rewards, in addition to the dynamics. We focus on only additive reward term, as we only need to estimate the reward in expectation, for which this structure of test functions is sufficient. The additional dependence on the context $x^1$ allows us to capture contextual RL setups [Hallak et al., 2015]. This assumption captures a wide range of structures including tabular, factored, linear and low-rank MDPs (see Sun et al. [2019] for further examples).

The bilinear structure of the factorization enables us to decouple the Bellman error. We begin with the case of finite action sets studied in Sun et al. [2019]. Let $p \circ^h q$ be a non-stationary policy which follows $\pi \sim p$ for $h - 1$ steps, and chooses $a^h \sim \pi'(\cdot|x^h)$ for $\pi' \sim q$.

**Proposition 3** *Under Assumption 3, suppose further that $|\mathcal{A}| = K$. Let us define $z_1 = x^1, z_2 = M$ and $\chi = \psi^h(M, x^1)$ in Definition 1. Then for any $\epsilon > 0$, we have*

$$\mathrm{dc}(\epsilon, p, \pi_{\mathrm{gen}}(h, p), 0.5) \leq \frac{4K}{\kappa^2} d_{\mathrm{eff}}\left(\psi^h, \frac{\kappa}{B_1}\epsilon\right), \quad \text{where } \pi_{\mathrm{gen}}(h, p) = p \circ^h \mathrm{Unif}(\mathcal{A}).$$

The proofs of Proposition 3 and all other results in this section are in Appendix E.

**Sample complexity under low witness rank and finite actions.** For ease of discussion, let $\dim(\psi^h) \leq d$ for all $h \in [H]$ and $|\mathcal{M}| < \infty$. Plugging Proposition 3 into Theorem 1 gives:

**Corollary 1** *Under conditions of Theorem 1, suppose further that Assumption 3 holds with $\dim(\psi^h) \leq d$ for all $h \in [H]$. Let the model class $\mathcal{M}$ an action space $\mathcal{A}$ have a finite cardinality with $|\mathcal{A}| = K$. Then* MOPS *satisfies* $\frac{1}{T}\sum_{t=1}^{T} V^\star(x_t^1) - V^{\pi_t}(x_t^1) \leq \mathcal{O}\left(\sqrt{\frac{H^2 d^2 K \ln |\mathcal{M}|}{\kappa^2 T}}\right)$.

With a standard online-to-batch conversion argument [Cesa-Bianchi et al., 2004], this implies a sample complexity bound to find an $\epsilon$-suboptimal policy of $\mathcal{O}\left(\frac{H^2 dK \ln |\mathcal{M}|}{\kappa^2 \epsilon^2}\right)$, when the contexts are i.i.d. from a distribution. This bound improves upon those of Sun et al. [2019],and Du et al. [2021], who require $\tilde{\mathcal{O}}\left(\frac{H^3 d^2 K}{\kappa^2 \epsilon^2} \ln \frac{T|\mathcal{M}||\mathcal{F}|}{\delta}\right)$ samples, where $\mathcal{F}$ is a discriminator class explicitly used in their algorithm, while we only use the discriminators implicitly in our analysis.

**Factored MDPs** Sun et al. [2019] show an exponential separation between sample complexity of model-based and model-free learning in factored MDPs [Boutilier et al., 1995] by controlling the error of each factor independently. The gap is demonstrated by choosing a discriminator class that measures the error separately on each factor in their algorithm. A similar adaptation of our approach to measure the likelihood of each factor separately in the setting of Proposition 3 allows our technique to handle factored MDPs.

Next, we further generalize this decoupling result to large action spaces by making a linear embedding assumption that can simultaneously capture all finite action problems, as well as some more general settings [Zhang, 2021, Zhang et al., 2021, Agarwal and Zhang, 2022].

**Assumption 4** (Linear embeddability of backup errors) *Let $\mathcal{F} = \{f(x, a, r, x') = r + g(x, a, x') : g \in \mathcal{G}\}$, with $g(x, a, x') \in [0, 1]$, be given. There exist (unknown) functions $\phi^h(x^h, a^h)$ and $w^h(M, f, x^h)$ such that $\forall f \in \mathcal{F}, M \in \mathcal{M}, h \in [H]$ and $x^h, a^h$:*

$$(P_M^h f)(x^h, a^h) - (P_\star^h f)(x^h, a^h) = \left\langle w^h(M, f, x^h), \phi^h(x^h, a^h) \right\rangle.$$

*We assume that $\|w^h(M, f, x^h)\|_2 \leq B_2$ for all $M, f, x^h$ and $h \in [H]$.*

Since the weights $w^h$ can depend on both the $f$ and $x^h$, for finite action problems it suffices to choose $\phi^h(x, a) = e_a$ and $w^h(M, f, x^h) = ((P_M^h f)(x^h, a) - (P_\star^h f)(x^h, a))_{a=1}^K$. For linear MDPs, the assumption holds with $\phi^h$ being the MDP features and $w^h$ being independent of $x^h$. A similar assumption on Bellman errors has recently been used in the analysis of model-free strategies for non-linear RL scaling to large action spaces [Zhang, 2021, Zhang et al., 2021, Agarwal and Zhang, 2022]. Note that Assumption 4 posits a pointwise factorization for each $x^h, a^h$ and for individual test functions $f \in \mathcal{F}$, and in general is not directly comparable to Assumption 3, which assumes a factorization of the average error in backups in the worst-case over all test functions. Assumption 3

allows us to decouple the distribution of $x^h$ from the model being evaluated, while Assumption 4 is needed to decouple the choice of actions $a^h$. Obtaining a decoupling as per Definition 2 requires using the two assumptions together.

We now state a general bound on the decoupling coefficient under Assumptions 3 and 4.

**Proposition 4** *Suppose Assumptions 3 and 4 hold. For $\phi^h$ in Definition 1, we define $z_1 = x^h$ and $z_2 = a^h$, with same choices for $\psi^h$ as Proposition 3. Then for any $\epsilon > 0$, we have*

$$\mathrm{dc}^h(\epsilon, p, \pi_{\mathrm{gen}}(h, p), 0.25) \leq 16\kappa^{-4} d_{\mathrm{eff}}(\psi^h, \epsilon_1)^2 d_{\mathrm{eff}}(\phi^h, \epsilon_2), \quad where\ \pi_{\mathrm{gen}}(h, p) = p \circ^h p$$

*for any $\epsilon_1, \epsilon_2 > 0$ satisfying $^{B_1 \epsilon_1}/\kappa + 2B_2 \epsilon_2 \sqrt{d_{\mathrm{eff}}(\psi^h, \epsilon_1)}/\kappa = \epsilon$.*

Compared with Proposition 3, we see that the exponent $\alpha$ changes to $0.25$ in Proposition 4. This happens because we now change the action choice at step $h$ to be from $\pi_{M'}$, where $M' \sim p$ independent of $M$. To carry out decoupling for this choice, we need to use both the factorizations in Assumptions 3 and 4, which introduces an additional Cauchy-Schwarz step. This change is necessary as no obvious exploration strategy like uniform exploration of Proposition 3 is available here.

**Sample complexity for low witness rank and (unknown) linear embedding.** Similar to Corollary 1, we can obtain a concrete result for this setting by combining Proposition 4 and Theorem 1.

**Corollary 2** *Under conditions of Theorem 1, suppose further that Assumptions 3 and 4 hold, and $|\mathcal{M}| < \infty$. For any $\epsilon_1, \epsilon_2 > 0$, let $d_{\mathrm{eff}}(\psi^h, \epsilon_1) \leq d_{\mathrm{eff}}(\psi, \epsilon_1)$ and $d_{\mathrm{eff}}(\phi^h, \epsilon_2) \leq d_{\mathrm{eff}}(\phi, \epsilon_2)$, for all $h \in [H]$. Then MOPS satisfies $\frac{1}{T} \sum_{t=1}^{T} V^\star(x_t^1) - V^{\pi_t}(x_t^1) \leq \mathcal{O}\left( \frac{H}{\kappa} \left( \frac{d_{\mathrm{eff}}(\psi, \epsilon_1)^2 d_{\mathrm{eff}}(\phi, \epsilon_2) \ln |\mathcal{M}|}{T} \right)^{1/4} + \left( \frac{B_1 \epsilon_1}{\kappa} + \frac{\sqrt{2B_2 \epsilon_2 d_{\mathrm{eff}}(\psi, \epsilon_1)}}{\kappa} \right) H \right).$*

When the maps $\psi^h, \phi^h$ are both finite dimensional with $\dim(\psi^h) \leq d_1$ and $\dim(\psi^h) \leq d_2$ for all $h$, MOPS enjoys a sample complexity of $\mathcal{O}\left( \frac{8 d_1^2 d_2 H^4 \ln |\mathcal{M}|}{\kappa^4 \epsilon^4} \right)$ in this setting. We are not aware of any prior methods for this setting. We note that similar models have been studied in a model-free framework in Agarwal and Zhang [2022], and that result obtains the same suboptimal $\epsilon^{-4}$ rate as shown here. The loss of rates in $T$ arises due to the worse exponent of $0.25$ in the decoupling bound of Proposition 4, which is due to the extra Cauchy-Schwarz step as mentioned earlier. For more general infinite dimensional cases, it is straightforward to develop results based on an exponential or polynomial spectral decay analogous to the prior model-free work.

## 6.2 $Q$-type decoupling and linear models

We now give examples of two other structural assumptions, where the decoupling holds pointwise for all $x$ and not just in expectation. As this is somewhat analogous to similar phenomena in $Q$-type Bellman rank [Jin et al., 2021], we call such results $Q$-type decouplings. We begin with the first assumption which applies to linear MDPs as well as certain models in continuous control, including Linear Quadratic Regulator (LQR) and the KNR model.

**Assumption 5** ($Q$-type witness factorization) *Let $\mathcal{F}$ be a function class such that $f(x, a, r, x') = r + g(x, a, x')$ for $g \in \mathcal{G}$, with $g(x, a, x') \in [0, 1]$, for all $x, a, x'$. Then there exist maps $\psi^h(x^h, a^h)$ and $u^h(M, f)$ and a constant $\kappa > 0$, such that for any $h, x^h, a^h, f \in \mathcal{F}$ and $M \in \mathcal{M}$:*

$$|(P_M f)(x^h, a^h) - (P^\star f)(x^h, a^h)| \geq \xi(M, f, x^h, a^h), \quad \sup_{f \in \mathcal{F}} \xi(M, f, x^h, a^h) \geq \kappa \mathcal{E}_B(M, x^h, a^h),$$

*with $\xi(M, f, x^h, a^h) = \left| \langle \psi^h(x^h, a^h), u^h(M, f) \rangle \right|$. We assume $\|u^h(M, f)\|_2 \leq B_1$ for all $M, f, h$.*

Assumption 5 is clearly satisfied by a linear MDP when $\phi^h$ are the linear MDP features and $\mathcal{F}$ is any arbitrary function class. We show in Appendix D.1 that this assumption also includes the Kernelized Non-linear Regulator (KNR), introduced in Kakade et al. [2020] as a model for continuous control that generalizes LQRs to handle some non-linearity. In a KNR, the dynamics follow $x^{h+1} = W^\star \varphi(x^h, a^h) + \epsilon$, with $\epsilon \sim \mathcal{N}(0, \sigma^2 I)$, and the features $\varphi(x^h, a^h)$ are known and lie in an RKHS. In this case, for an appropriate class $\mathcal{F}$, we again get Assumption 5 with features $\phi^h = \varphi$ from the KNR dynamics.

Under this assumption, we get an immediate decoupling result, with the proof in Appendix F.

**Proposition 5** *Under Assumption 5, let define $z_2 = (x^h, a^h)$ and $\chi = \psi^h(x^h, a^h)$ in Definition 1. Then for any $\epsilon > 0$, we have*

$$\mathrm{dc}(\epsilon, p, \pi_{\mathrm{gen}}(h, p), 0.5) \leq \frac{4}{\kappa^2} d_{\mathrm{eff}}\left(\psi^h, \frac{\kappa}{B_1}\epsilon\right), \quad where\ \pi_{\mathrm{gen}}(h, p) = \pi_M\ with\ M \sim p.$$

Here we see that the decoupling coefficient scales with the effective feature dimension, which now simultaneously captures the exploration complexity over both states and actions, consistent with existing results for $Q$-type settings such as linear MDPs.

**Sample complexity for the KNR model.** We now instantiate a concrete corollary of Theorem 1 for the KNR model, under the assumption that $x^h \in \mathbb{R}^{d_{\mathcal{X}}}$, $\varphi(x^h, a^h) \in \mathbb{R}^{d_\varphi}$ and $\|\varphi(x^h, a^h)\|_2 \leq B$ for all $x^h, a^h$ and $h \in [H]$. A similar result also holds for all problems where Assumption 5 holds, but we state a concrete result for KNRs to illustrate the handling of infinite model classes.

**Corollary 3** *Under conditions of Theorem 1, suppose further that we apply* MOPS *to the KNR model with the model class $\mathcal{M}_{KNR} = \{W \in \mathbb{R}^{d_{\mathcal{X}} \times d_\varphi} : \|W\|_2 \leq R\}$. The* MOPS *satisfies:*
$$\frac{1}{T}\sum_{t=1}^T V^\star(x_t^1) - V^{\pi_t}(x_t^1) \leq \mathcal{O}\left(H\sqrt{\frac{d_\varphi^2 d_{\mathcal{X}} \log(R\sqrt{d_\varphi}BHT/\sigma)}{T\sigma^2}}\right).$$

Structurally, the result is a bit similar to Corollary 1, except that there is no action set dependence any more, since the feature dimension captures both state and action space complexities in the $Q$-type setting as remarked before. It is proved by using the bound on $\omega(\alpha, p_0)$ in Lemma 11 in Appendix D.1 and the decoupling coefficient bound in Proposition 7 in Appendix F with $d_{\mathrm{eff}} = d_\varphi$.

We also do not make a finite model space assumption in this result, as mentioned earlier. To apply Corollary 3 to this setting, we bound $\omega(T, p_0)$ in Lemma 11 in Appendix D.1. We notice that Corollary 3 has a slightly inferior $d_\phi^2 d_{\mathcal{X}}$ dimension dependence compared to the $d_\phi(d_{\mathcal{X}} + d_\phi)$ scaling in Kakade et al. [2020]. It is possible to bridge this gap by a direct analysis of the algorithm in this case, with similar arguments as the KNR paper, but our decoupling argument loses an extra dimension factor. Note that it is unclear how to cast the broader setting of Assumption 5 in the frameworks of Bilinear classes or DEC.

**Linear Mixture MDPs.** A slightly different $Q$-type factorization assumption which includes linear mixture MDPs [Modi et al., 2020, Ayoub et al., 2020], also amenable to decoupling, is discussed in Appendix D.2. For this model, we get an error bound of $\mathcal{O}\left(H\sqrt{\frac{d_\phi \omega(3HT, p_0)}{T\kappa^2}}\right)$. Given our assumption on value functions normalization by 1, this suggests a suboptimal scaling in $H$ factors [Zhou et al., 2021], because our algorithm uses samples from a randomly chosen time step only and our analysis does not currently leverage the Bellman property of variance, which is crucial to a sharper analysis. While addressing the former under the $Q$-type assumptions is easy, improving the latter for general RL settings is an exciting research direction.

# 7 Conclusion

This paper proposes a general algorithmic and statistical framework for model-based RL bassd on optimistic posterior sampling. The development yields state-of-the-art sample complexity results under several structural assumptions. Our techniques are also amenable to practical adaptations, as opposed to some prior attempts relying on complicated constrained optimization objectives that may be difficult to solve. Empirical evaluation of the proposed algorithms would be interesting for future work. As another future direction, our analysis (and that of others in the rich observation setting) does not leverage the Bellman property of variance, which is essential for sharp horizon dependence in tabular [Azar et al., 2017] and some linear settings [Zhou et al., 2021]. Extending these ideas to general non-linear function approximation is an important direction for future work. More generally, understanding if the guarantees can be more adaptive on a per-instance basis, instead of worst-case, is critical for making this theory more practical.

# Acknowledgements

The authors thank Wen Sun for giving feedback on an early draft of this work.

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
