# A    Proof of Proposition 1

By Lemma 1, we have

$$\mathop{\mathbb{E}}_{M \sim p_t} \mathbb{E}[V^\star(x_t^1) - V^{\pi_M}(x_t^1)] = \mathop{\mathbb{E}}_{M \sim p_t} \mathbb{E} \left[ \sum_{h=1}^{H} \underbrace{\mathop{\mathbb{E}}_{x^h, a^h \sim \pi_M | x_t^1} \mathcal{E}_B(M, x^h, a^h)}_{\mathcal{T}} - \Delta V_M(x_t^1) \right]. \quad (4)$$

We now focus on $\mathcal{T}$, which can be bounded using Definition 2 as follows,

$$\mathbb{E} \sum_{h=1}^{H} \mathop{\mathbb{E}}_{M \sim p_t} \mathop{\mathbb{E}}_{x^h, a^h \sim \pi_M | x_t^1} \mathcal{E}_B(M, x^h, a^h)$$

$$= H \mathbb{E} \mathop{\mathbb{E}}_{h_t} \mathop{\mathbb{E}}_{M \sim p_t} \mathop{\mathbb{E}}_{(x^{h_t}, a^{h_t}) \sim \pi_M(\cdot | x_t^1)} \mathcal{E}_B(M, x_t^{h_t}, a_t^{h_t})$$

$$\leq H \mathbb{E} \mathop{\mathbb{E}}_{h_t} \left[ \left( \mathrm{dc}^{h_t}(\epsilon, \alpha) \mathop{\mathbb{E}}_{M \sim p_t} \mathop{\mathbb{E}}_{(x_t^{h_t}, a_t^{h_t}) \sim \pi_t(\cdot | x_t^1)} \ell^h(M, x_t^{h_t}, a_t^{h_t}) \right)^\alpha + \epsilon \right]$$

$$\leq H \mathbb{E} \mathop{\mathbb{E}}_{h_t} \left[ \epsilon + (1 - \alpha)(\mu/\alpha)^{-\alpha/(1-\alpha)} \mathrm{dc}^{h_t}(\epsilon, \alpha)^{\alpha/(1-\alpha)} + \mu \mathop{\mathbb{E}}_{M \sim p_t} \mathop{\mathbb{E}} \ell^{h_t}(M, x_t^{h_t}, a_t^{h_t}) \right]$$

$$= \epsilon H + H(1 - \alpha)(\mu/\alpha)^{-\alpha/(1-\alpha)} \mathrm{dc}(\epsilon, \alpha)^{\alpha/(1-\alpha)} + H \mathbb{E} \mathop{\mathbb{E}}_{h_t} \left[ \mu \mathop{\mathbb{E}}_{M \sim p_t} \mathop{\mathbb{E}} \ell^{h_t}(M, x_t^{h_t}, a_t^{h_t}) \right]. \quad (5)$$

The first inequality used Definition 2. The second inequality used the following inequality:

$$(ab)^\alpha = \inf_{\mu > 0} \left[ \mu b + (1 - \alpha)(\mu/\alpha)^{-\alpha/(1-\alpha)} a^{\alpha/(1-\alpha)} \right].$$

Plugging (5) into (4), we obtain the bound.

# B    Analysis of online learning

We need some additional notation for this part of the analysis. Let us define

$$\Delta L_t^h(M) = L_t^h(M) - L_t^h(M_\star).$$

We also define

$$\xi_1^h(M, x_t^h, a_t^h, r_t^h) = -3\eta[(R_M(x_t^h, a_t^h) - r_t^h)^2 - (R_\star(x_t^h, a_t^h) - r_t^h)^2]$$

$$\xi_2^h(M, x_t^h, a_t^h, x_t^{h+1}) = 3\eta' \ln \frac{P_M(x_t^{h+1} \mid x_t^h, a_t^h)}{P_\star(x_t^{h+1} \mid x_t^h, a_t^h)}. \quad (6)$$

With this notation, we first perform some simple manipulations to the posterior distribution, before relating it to the Hellinger distance that arises in our regret analysis.

**Lemma 3.**

$$\sum_{t=1}^{T} \mathop{\mathbb{E}}_{S_t} \ln \mathop{\mathbb{E}}_{M \sim p(M | S_{t-1})} \exp \left( \gamma \Delta V_M(x_t^1) + \Delta L_t^{h_t}(M) \right)$$

$$= \mathop{\mathbb{E}}_{S_T} \ln \mathop{\mathbb{E}}_{M \sim p_0} \exp \left( \sum_{t=1}^{T} (\gamma \Delta V_M(x_t^1) + \Delta L_t^{h_t}(M)) \right).$$

*Proof.* We have

$$\mathbb{E}_{S_t} \ln \mathbb{E}_{M\sim p(M|S_{t-1})} \exp\left(\gamma\Delta V_M(x_t^1) + \Delta L_t^{h_t}(M)\right)$$

$$= \mathbb{E}_{S_t} \ln \mathbb{E}_{M\sim p_0} \frac{\exp\left(\sum_{s=1}^{t-1} \gamma V_M(x_s^1) + L_s^{h_s}(M)\right)}{\mathbb{E}_{M'\sim p_0} \exp\left(\sum_{s=1}^{t-1} \gamma V_{M'}(x_s^1) + L_s^{h_s}(M')\right)} \exp\left(\gamma\Delta V_M(x_t^1) + \Delta L_t^{h_t}(M)\right)$$

$$= \mathbb{E}_{S_t} \ln \mathbb{E}_{M\sim p_0} \frac{\exp\left(\sum_{s=1}^{t} \gamma V_M(x_s^1) + L_s^{h_s}(M)\right)}{\mathbb{E}_{M'\sim p_0} \exp\left(\sum_{s=1}^{t-1} \gamma V_{M'}(x_s^1) + L_s^{h_s}(M')\right)} \exp\left(-\gamma V^\star(x_t^1) - L_t^{h_t}(M_\star)\right)$$

$$= \mathbb{E}_{S_t} \ln \mathbb{E}_{M\sim p_0} \exp\left(\sum_{s=1}^{t} \gamma\Delta V_M(x_s^1) + \Delta L_s^{h_s}(M)\right) - \mathbb{E}_{S_{t-1}} \ln \mathbb{E}_{M\sim p_0} \exp\left(\sum_{s=1}^{t-1} \gamma\Delta V_M(x_s^1) + \Delta L_s^{h_s}(M)\right).$$

Here the first equality follows from our definition of the posterior in Line 4 of Algorithm 1 and the last equality multiplies and divides the ratio by $\exp\left(-\sum_{s=1}^{t-1} \gamma V^\star(x_s^1) - L_s^{h_s}(M_\star)\right)$. Now adding terms from $t=1$ to $T$ and telescoping completes the proof. $\qquad\square$

Next we separate the terms corresponding to the optimistic value function, reward and dynamics posteriors. Doing so, gives the following lemma.

**Lemma 4.** Let $\mathbb{E}_t$ denotes the expectation over $(x_t, a_t, r_t)$ conditioned on $S_{t-1}$. We have

$$\frac{1}{3}\ln\mathbb{E}_t \mathbb{E}_{M\sim p(M|S_{t-1})} \exp\left(3\gamma\Delta V_M(x_t^1)\right)$$

$$+ \frac{1}{3}\ln\mathbb{E}_t \mathbb{E}_{M\sim p(M|S_{t-1})} \mathbb{E}_{r_t^{h_t}|x_t^{h_t},a_t^{h_t}} \exp\left(-3\eta((R_M(x_t^{h_t}, a_t^{h_t}) - r_t^{h_t})^2 - (R_\star(x_t^{h_t}, a_t^{h_t}) - r_t^{h_t})^2)\right)$$

$$+ \frac{1}{3}\ln\mathbb{E}_t \mathbb{E}_{M\sim p(M|S_{t-1})} \mathbb{E}_{x_t^{h_t+1}|x_t^{h_t},a_t^{h_t}} \exp\left(3\eta' \ln \frac{P_M(x_t^{h_t+1} \mid x_t^{h_t}, a_t^{h_t})}{P_\star(x_t^{h_t+1} \mid x_t^{h_t}, a_t^{h_t})}\right)$$

$$\geq \mathbb{E}_t \ln \mathbb{E}_{M\sim p(M|S_{t-1})} \exp\left(\gamma\Delta V_M(x_t^1) + \Delta L_t^{h_t}(M)\right)$$

*Proof.* Using (6), we can write

$$\mathbb{E}_t \ln \mathbb{E}_{M\sim p(\cdot|S_{t-1})} \exp\left(\gamma\Delta V_M(x_t^1) + \Delta L_t^{h_t}(M)\right)$$

$$= \mathbb{E}_t \ln \mathbb{E}_{M\sim p(\cdot|S_{t-1})} \exp\left(\frac{1}{3}(3\gamma\Delta V_M(x_t^1)) + \frac{1}{3}(\xi_{1,t}^{h_t}(M, x_t^h, a_t^h, r_t^h) + \frac{1}{3}\xi_{2,t}^{h_t}(M, x_t^h, a_t^h, x_t^{h+1})\right)$$

$$\leq \ln \mathbb{E}_t \mathbb{E}_{M\sim p(\cdot|S_{t-1})} \exp\left(\frac{1}{3}(3\gamma\Delta V_M(x_t^1)) + \frac{1}{3}(\xi_{1,t}^{h_t}(M, x_t^h, a_t^h, r_t^h) + \frac{1}{3}\xi_{2,t}^{h_t}(M, x_t^h, a_t^h, x_t^{h+1})\right)$$

$$\leq \frac{1}{3}\ln\mathbb{E}_t \mathbb{E}_{M\sim p(\cdot|S_{t-1})} \exp\left(3\gamma\Delta V_M(x_t^1)\right) + \frac{1}{3}\ln\mathbb{E}_t \mathbb{E}_{M\sim p(\cdot|S_{t-1})} \exp\left(\xi_{1,t}^{h_t}(M, x_t^h, a_t^h, r_t^h)\right)$$

$$+ \frac{1}{3}\ln\mathbb{E}_t \mathbb{E}_{M\sim p(\cdot|S_{t-1})} \exp\left(\xi_{2,t}^{h_t}(M, x_t^h, a_t^h, x_t^{h+1})\right).$$

Here the inequalities follow from Jensen's inequality. This yields the conclusion of the lemma. $\quad\square$

We now relate the last two terms in Lemma 4 to squared loss regret in rewards and expected Hellinger loss of the posterior distribution. It would be helpful to recall the following moment generating function bound for bounded random variables, from the proof of Hoeffding's inequality.

**Lemma 5** (Hoeffding's inequality)**.** Let $Z$ be a zero-mean random variable such that $\max[Z] \leq a + \min[Z]$. Then $\mathbb{E}[\exp(cZ)] \leq \exp(a^2c^2/8)$.

**Lemma 6.** Suppose $\eta \le 1/3$. We have

$$\frac{1.5\eta}{\sqrt{e}} \, \mathbb{E}_t \, \mathbb{E}_{M \sim p(\cdot|S_{t-1})} (R_M(x_t^{h_t}, a_t^{h_t}) - R_\star(x_t^{h_t}, a_t^{h_t}))^2$$

$$\le - \ln \mathbb{E}_t \, \mathbb{E}_{M \sim p(M|S_{t-1})} \, \mathbb{E}_{r_t^{h_t}|x_t^{h_t}, a_t^{h_t}} \exp\left(-3\eta((R_M(x_t^{h_t}, a_t^{h_t}) - r_t^{h_t})^2 - (R_\star(x_t^{h_t}, a_t^{h_t}) - r_t^{h_t})^2)\right).$$

*Proof.* Using Jensen's inequality, we see that

$$\ln \mathbb{E}_t \, \mathbb{E}_{M \sim p(\cdot|S_{t-1})} \, \mathbb{E}_{r_t^{h_t}|x_t^{h_t}, a_t^{h_t}} \exp\left(-3\eta((R_M(x_t^{h_t}, a_t^{h_t}) - r_t^{h_t})^2 - (R_\star(x_t^{h_t}, a_t^{h_t}) - r_t^{h_t})^2)\right)$$

$$\le \mathbb{E}_t \, \mathbb{E}_{M \sim p(\cdot|S_{t-1})} \, \mathbb{E}_{r_t^{h_t}|x_t^{h_t}, a_t^{h_t}} \exp\left(-3\eta((R_M(x_t^{h_t}, a_t^{h_t}) - r_t^{h_t})^2 - (R_\star(x_t^{h_t}, a_t^{h_t}) - r_t^{h_t})^2)\right) - 1$$

$$= \mathbb{E}_t \, \mathbb{E}_{M \sim p(\cdot|S_{t-1})} \exp\left(-3\eta(R_M(x_t^{h_t}, a_t^{h_t}) - R_\star(x_t^{h_t}, a_t^{h_t}))^2\right)$$

$$\cdot \mathbb{E}_{r_t^{h_t}|x_t^{h_t}, a_t^{h_t}} \exp\left(6\eta(R_\star(x_t^{h_t}, a_t^{h_t}) - r_t^{h_t})(R_\star(x_t^{h_t}, a_t^{h_t}) - R_M(x_t^{h_t}, a_t^{h_t}))\right) - 1$$

$$\le \mathbb{E}_t \, \mathbb{E}_{M \sim p(\cdot|S_{t-1})} \exp\left(-(3\eta - 4.5\eta^2)(R_M(x_t^{h_t}, a_t^{h_t}) - R_\star(x_t^{h_t}, a_t^{h_t}))^2\right) - 1 \qquad \text{(Lemma 5)}$$

$$\le \mathbb{E}_t \, \mathbb{E}_{M \sim p(\cdot|S_{t-1})} \exp\left(-1.5\eta(R_M(x_t^{h_t}, a_t^{h_t}) - R_\star(x_t^{h_t}, a_t^{h_t}))^2\right) - 1 \qquad (\eta \le 1/3)$$

$$\le - \frac{1.5\eta}{\sqrt{e}} \, \mathbb{E}_t \, \mathbb{E}_{M \sim p(\cdot|S_{t-1})} (R_M(x_t^{h_t}, a_t^{h_t}) - R_\star(x_t^{h_t}, a_t^{h_t}))^2.$$

$$\text{(} 1.5\eta \le 0.5 \text{ and } e^x \le 1 + x/\sqrt{e} \text{ for } x \in [-0.5, 0]\text{)}$$

$\square$

Next we give a similar result regarding the dynamics suboptimality. This bound is analogous to the techniques used in the analysis of maximum likelihood estimation in Hellinger distance [see e.g. Zhang, 2006, van de Geer, 2000].

**Lemma 7.** Suppose $3\eta' = 0.5$. We have

$$\frac{1}{2} \, \mathbb{E}_t \, \mathbb{E}_{M \sim p(\cdot|S_{t-1})} D_H(P_M(x_t^{h_t+1}|x_t^{h_t}, a_t^{h_t}), P_\star(x_t^{h_t+1}|x_t^{h_t}, a_t^{h_t}))^2$$

$$\le - \ln \mathbb{E}_t \, \mathbb{E}_{M \sim p(M|S_{t-1})} \, \mathbb{E}_{x_t^{h_t+1}|x_t^{h_t}, a_t^{h_t}} \exp\left(3\eta' \ln \frac{P_M(x_t^{h_t+1} \mid x_t^{h_t}, a_t^{h_t})}{P_\star(x_t^{h_t+1} \mid x_t^{h_t}, a_t^{h_t})}\right).$$

*Proof.* Using Jensen's inequality, we see that

$$\ln \mathbb{E}_t \, \mathbb{E}_{M \sim p(\cdot|S_{t-1})} \, \mathbb{E}_{x_t^{h_t+1}|x_t^{h_t}, a_t^{h_t}} \exp\left(3\eta' \ln \frac{P_M(x_t^{h_t+1}|x_t^{h_t}, a_t^{h_t})}{P_\star(x_t^{h_t+1}|x_t^{h_t}, a_t^{h_t})}\right)$$

$$\le \mathbb{E}_t \, \mathbb{E}_{M \sim p(\cdot|S_{t-1})} \, \mathbb{E}_{x_t^{h_t+1}|x_t^{h_t}, a_t^{h_t}} \exp\left(3\eta' \ln \frac{P_M(x_t^{h_t+1}|x_t^{h_t}, a_t^{h_t})}{P_\star(x_t^{h_t+1}|x_t^{h_t}, a_t^{h_t})}\right) - 1$$

$$= \mathbb{E}_t \, \mathbb{E}_{M \sim p(\cdot|S_{t-1})} \, \mathbb{E}_{x_t^{h_t+1}|x_t^{h_t}, a_t^{h_t}} \int \sqrt{dP_M(x_t^{h_t+1}|x_t^{h_t}, a_t^{h_t})dP_\star(x_t^{h_t+1}|x_t^{h_t}, a_t^{h_t})} - 1 \qquad (3\eta' = 0.5)$$

$$= - \frac{1}{2} \, \mathbb{E}_t \, \mathbb{E}_{M \sim p(\cdot|S_{t-1})} D_H(P_M(x_t^{h_t+1}|x_t^{h_t}, a_t^{h_t}), P_\star(x_t^{h_t+1}|x_t^{h_t}, a_t^{h_t}))^2.$$

$\square$

Finally, we give a bound for the optimism term.

**Lemma 8.** We have

$$-3\gamma \mathop{\mathbb{E}}_{t} \mathop{\mathbb{E}}_{M\sim p(\cdot|S_{t-1})} \Delta V_M(x_t^1) \leq -\ln \mathop{\mathbb{E}}_{t} \mathop{\mathbb{E}}_{M\sim p(\cdot|S_{t-1})} \exp(3\gamma\Delta V_M(x_t^1)) + \frac{9}{2}\gamma^2.$$

*Proof.* We have

$$\ln \mathop{\mathbb{E}}_{t} \mathop{\mathbb{E}}_{M\sim p(\cdot|S_{t-1})} \exp(3\gamma\Delta V_M(x_t^1))$$

$$=\ln \mathop{\mathbb{E}}_{t} \mathop{\mathbb{E}}_{M\sim p(\cdot|S_{t-1})} \exp(3\gamma(\Delta V_M(x_t^1) - \mathop{\mathbb{E}}_{t} \mathop{\mathbb{E}}_{M\sim p(\cdot|S_{t-1})} \Delta V_M(x_t^1))) + 3\gamma \mathop{\mathbb{E}}_{t} \mathop{\mathbb{E}}_{M\sim p(\cdot|S_{t-1})} \Delta V_M(x_t^1)$$

$$\leq 3\gamma \mathop{\mathbb{E}}_{t} \mathop{\mathbb{E}}_{M\sim p(\cdot|S_{t-1})} \Delta V_M(x_t^1) + \frac{9}{2}\gamma^2,$$

where the last inequality used $\Delta V_M(x_t^1) \in [-1, 1]$, and Lemma 5. $\square$

We now give a useful lemma in relating the value suboptimality to the model and reward errors, which will be used to complete our online learning analysis.

**Lemma 9.** Recall the set $\mathcal{M}(\epsilon)$ in Definition 3. For any $M \in \mathcal{M}(\epsilon)$, we have

$$\sup_{x^1} |\Delta V_M(x^1)| \leq 3H\epsilon.$$

*Proof.* The proof of Lemma 1 gives that for any two models, $M$ and $M'$, we have

$$V_M(x^1) - V_{M'}(x^1) \leq V_M(x^1) - V_{M'}^{\pi_M}(x^1)$$

$$=\sum_{h=1}^{H} \mathbb{E}_{x^h,a^h\sim\pi_M}^{M'} \left[Q_M^h(x^h, a^h) - (P_{M'}^h[r + V_M^{h+1}])(x^h, a^h)\right]$$

$$=\sum_{h=1}^{H} \mathbb{E}_{x^h,a^h\sim\pi_M}^{M'} \left[R_M^h(x^h, a^h) - R_{M'}^h(x^h, a^h) + (P_M^h V_M^{h+1})(x^h, a^h) - (P_{M'}^h V_M^{h+1}])(x^h, a^h)\right]$$

$$\leq H \sup_{h,x^h,a^h} \left[\sqrt{(R_M^h(x^h, a^h) - R_{M'}^h(x^h, a^h))^2} + 2\mathrm{TV}(P_M^h(\cdot|x^h, a^h), P_{M'}^h(\cdot|x^h, a^h))\right]$$

$$\leq H \sup_{h,x^h,a^h} \left[\sqrt{(R_M^h(x^h, a^h) - R_{M'}^h(x^h, a^h))^2} + \sqrt{2\mathrm{KL}(P_M^h(\cdot|x^h, a^h)||P_{M'}^h(\cdot|x^h, a^h))}\right],$$

where the last inequality follows from Pinsker's inequality.

Since the bound up to the TV error term is symmetric in $M$ and $M'$, the same bound also holds for the absolute value of the value differences. Now applying the bound with $M' = M_\star$, we see that for any model $M \in \mathcal{M}(\epsilon)$, we have

$$\sup_{x^1} |\Delta V_M(x^1)| \leq H \sup_{h,x^h,a^h} \left[\sqrt{(R_M^h(x^h, a^h) - R_\star^h(x^h, a^h))^2} + \sqrt{2\mathrm{KL}(P_\star^h(\cdot|x^h, a^h)||P_M^h(\cdot|x^h, a^h))}\right]$$

$$\leq H(1 + \sqrt{2})\epsilon \leq 3H\epsilon,$$

where the inequality holds since each of the reward errors and the KL divergence term is individually bounded by $\epsilon^2$. $\square$

We complete our online learning analysis by bounding the log-partition function used in Lemma 3.

**Lemma 10.** For any $\epsilon \leq 3H$, we have

$$-\mathop{\mathbb{E}}_{S_T} \ln \mathop{\mathbb{E}}_{M\sim p_0} \exp\left(\sum_{t=1}^{T}(\gamma\Delta V_M(x_t^1) + \Delta L_t^{h_t}(M))\right) \leq \omega(3(\gamma + \eta + \eta')HT, p_0).$$

*Proof.* Given any $\epsilon > 0$, consider $\mathcal{M}(\epsilon)$ defined in Definition 3, we define $p_\epsilon$ as follows

$$p_\epsilon(M) = \frac{p_0(M)I(M \in \mathcal{M}(\epsilon))}{p_0(\mathcal{M}(\epsilon))}.$$

Let $p_t = p(f|S_{t-1})$ be the sampling distribution used at round $t$ in Algorithm 1. Note that

$$p_t = \operatorname*{argmin}_{p \in \Delta(\mathcal{M})} \sum_{s=1}^{t-1} \mathop{\mathbb{E}}_{M \sim p} \left( -\gamma V_M(x_s^1) - L_s^{h_s}(M) \right) + \mathrm{KL}(p||p_0),$$

We thus obtain that

$$\mathop{\mathbb{E}}_{S_T} \ln \mathop{\mathbb{E}}_{M \sim p_0} \exp\left( \sum_{t=1}^{T} (\gamma \Delta V_M(x_t^1) + \Delta L_t^{h_t}(M)) \right)$$

$$= \mathop{\mathbb{E}}_{S_T} \left[ \ln \mathop{\mathbb{E}}_{M \sim p_0} \exp\left( \sum_{t=1}^{T} (\gamma V_M(x_t^1) + L_t^{h_t}(M)) \right) - \sum_{t=1}^{T} (\gamma V^\star(x_t^1) - L_t^{h_t}(M_\star)) \right]$$

$$= \mathop{\mathbb{E}}_{S_T} \left[ \mathop{\mathbb{E}}_{M \sim p_T} \sum_{t=1}^{T} \left( \gamma V_M(x_t^1) + L_t^{h_t}(M) \right) - \sum_{t=1}^{T} (\gamma V^\star(x_t^1) - L_t^{h_t}(M_\star)) \right] - \mathrm{KL}(p_{T+1}||p_0)$$

$$\geq \mathop{\mathbb{E}}_{S_T} \left[ \mathop{\mathbb{E}}_{M \sim p_\epsilon} \sum_{t=1}^{T} \left( \gamma V_M(x_t^1) + L_t^{h_t}(M) \right) - \sum_{t=1}^{T} (\gamma V^\star(x_t^1) - L_t^{h_t}(M_\star)) \right] - \mathrm{KL}(p_\epsilon||p_0)$$

$$= \mathop{\mathbb{E}}_{S_T} \left[ \mathop{\mathbb{E}}_{M \sim p_\epsilon} \sum_{t=1}^{T} \left( \gamma \Delta V_M(x_t^1) - \eta(R_M^{h_t}(x_t^{h_t}, a_t^{h_t}) - R_{M_\star}^{h_t}(x_t^{h_t}, a_t^{h_t}))^2 - \eta' \mathrm{KL}(P_{M_\star}^{h_t}(\cdot|x_t^{h_t}, a_t^{h_t})||P_M^{h_t}(\cdot|x_t^{h_t}, a_t^{h_t})) \right) \right]$$

$$+ \ln p_0(\mathcal{M}(\epsilon))$$

$$\geq \mathop{\mathbb{E}}_{S_T} \left[ \mathop{\mathbb{E}}_{M \sim p_\epsilon} \sum_{t=1}^{T} - \left( 3H\gamma\epsilon + \eta\epsilon^2 + \eta'\epsilon^2 \right) \right] + \ln p_0(\mathcal{M}(\epsilon)) \geq -3(\gamma + \eta + \eta')\epsilon HT + \ln p_0(\mathcal{M}(\epsilon)),$$

where the second to last equality can be obtained by taking conditional expectation of $\mathbb{E}_{M_\star}[\cdot|x_t^{h_t}, a_t^{h_t}]$. The last inequality holds due to the definitions of $\mathcal{M}(\epsilon)$. The conclusion follows by taking inf over $\epsilon$ and using the notation $\omega$ from Definition 3. $\square$

We are now ready to prove Proposition 2.

*Proof of Proposition 2.* Noting that $0.9\eta \leq 1.5\eta/\sqrt{e} \leq 1/2$, we obtain

$$3\gamma \mathop{\mathbb{E}}_{t} \mathop{\mathbb{E}}_{M \sim p(\cdot|S_{t-1})} \left[ \frac{0.3\eta}{\gamma} \ell^{h_t}(M, x_t^{h_t}, a_t^{h_t}) - \Delta V_M(x_t^1) \right]$$

$$\leq \frac{1.5\eta}{\sqrt{e}} \mathop{\mathbb{E}}_{t} \mathop{\mathbb{E}}_{M \sim p(\cdot|S_{t-1})} (R_M(x_t^{h_t}, a_t^{h_t}) - R_\star(x_t^{h_t}, a_t^{h_t}))^2$$

$$\frac{1}{2} \mathop{\mathbb{E}}_{t} \mathop{\mathbb{E}}_{M \sim p(\cdot|S_{t-1})} D_H(P_M(x_t^{h_t+1}|x_t^{h_t}, a_t^{h_t}), P_\star(x_t^{h_t+1}|x_t^{h_t}, a_t^{h_t}))^2$$

$$- 3\gamma \mathop{\mathbb{E}}_{t} \mathop{\mathbb{E}}_{M \sim p(\cdot|S_{t-1})} \Delta V_M(x_t^1)$$

$$\leq - \ln \mathop{\mathbb{E}}_{t} \mathop{\mathbb{E}}_{M \sim p(M|S_{t-1})} \mathop{\mathbb{E}}_{r_t^{h_t}|x_t^{h_t}, a_t^{h_t}} \exp\left( -3\eta((R_M(x_t^{h_t}, a_t^{h_t}) - r_t^{h_t})^2 - (R_\star(x_t^{h_t}, a_t^{h_t}) - r_t^{h_t})^2) \right)$$
(Lemma 6)

$$- \ln \mathop{\mathbb{E}}_{t} \mathop{\mathbb{E}}_{M \sim p(M|S_{t-1})} \mathop{\mathbb{E}}_{x_t^{h_t+1}|x_t^{h_t}, a_t^{h_t}} \exp\left( 3\eta' \ln \frac{P_M(x_t^{h_t+1} \mid x_t^{h_t}, a_t^{h_t})}{P_\star(x_t^{h_t+1} \mid x_t^{h_t}, a_t^{h_t})} \right)$$
(Lemma 7)

$$- \ln \mathop{\mathbb{E}}_{t} \mathop{\mathbb{E}}_{M \sim p(\cdot|S_{t-1})} \exp(3\gamma \Delta V_M(x_t^1)) + \frac{9}{2}\gamma^2$$
(Lemma 8)

$$\leq - 3 \mathop{\mathbb{E}}_{t} \ln \mathop{\mathbb{E}}_{M \sim p(M|S_{t-1})} \exp\left( \gamma \Delta V_M(x_t^1) + \Delta L_t^{h_t}(M) \right) + \frac{9}{2}\gamma^2.$$
(Lemma 4)

It follows that

$$3\gamma \sum_{t=1}^{T} \left[ \frac{0.3\eta}{\gamma} \ell^{h_t}(M, x_t^{h_t}, a_t^{h_t}) - \Delta V_M(x_t^1) \right]$$

$$\leq -3 \sum_{t=1}^{T} \mathbb{E}_t \ln \mathbb{E}_{M \sim p(M|S_{t-1})} \exp\left( \gamma \Delta V_M(x_t^1) + \Delta L_t^{h_t}(M) \right) + \frac{9}{2}\gamma^2 T$$

$$= -3 \mathbb{E}_{S_T} \ln \mathbb{E}_{M \sim p_0} \exp\left( \sum_{t=1}^{T} (\gamma \Delta V_M(x_t^1) + \Delta L_t^{h_t}(M)) \right) + \frac{9}{2}\gamma^2 T \qquad \text{(Lemma 3)}$$

$$\leq 3\omega(3(\gamma + \eta + \eta')HT, p_0) + \frac{9}{2}\gamma^2 T. \qquad \text{(Lemma 10)}$$

Since $\eta + \eta' + \gamma < 1$, we the desired result. $\qquad \square$

## C  Proof of Lemma 2

The finite case is straightforward since choosing $p_0$ to be uniform over $\mathcal{M}$, we have for any $\epsilon \geq 0$ and $\alpha \geq 0$:

$$\alpha\epsilon - \ln p_0(\mathcal{M}(\epsilon)) \leq \alpha\epsilon + \ln |\mathcal{M}|,$$

since $M^\star$ is always contained in $\mathcal{M}(\epsilon)$ for all $\epsilon \geq 0$. This proves the first claim.

For the second claim, let $\mathcal{C}(\xi)$ be an $\ell_\infty$ cover of $\mathcal{M}$ at a radius $\xi$ under the metric $\ell^h$ for each level $h$, so that for any $M \in \mathcal{M}$, there is $M' \in \mathcal{C}(\xi)$ satisfying:

$$\sup_{h \in [H]} \sup_{x^h, a^h} |\ell^h(M, x^h, a^h) - \ell^h(M', x^h, a^h)| \leq \xi.$$

In particular, let $\tilde{M} = (\tilde{P}, \tilde{R})$ be the closest element to $M_\star$ in the covering. Since $\ell^h(M_\star, x^h, a^h) = 0$ for all $h, x^h, a^h$, we have that $\sup_{h, x^h, a^h} \ell(\tilde{M}, x^h, a^h) \leq \xi$. Let us further define $\mathcal{C}_\nu(\xi) = \{ (\nu P_0 + (1-\nu)P_M, R_M) : M \in \mathcal{C}(\xi) \}$ for some parameter $\nu \in (0, 1)$, where $P_0$ is as defined in Lemma 2. It is easily seen that $\|dP^\star/dP\|_\infty \leq B/\nu$ for any $P = \nu P_0 + (1-\nu)P_M$ by the definition of $P_0$. Furthermore, letting $P' = \nu P_0 + (1-\nu)\tilde{P}$ and $M' = (\nu P_0 + (1-\nu)\tilde{P}, \tilde{R})$, we see that for any $h \in [H]$ and $x^h, a^h$:

$$\ell^h(M', x^h, a^h)$$
$$= (\tilde{R}(x^h, a^h) - R_\star(x^h, a^h))^2 + D_H(P'(\cdot|x^h, a^h), P_\star(\cdot|x^h, a^h))^2$$
$$= (\tilde{R}(x^h, a^h) - R_\star(x^h, a^h))^2 + 2 - 2 \mathbb{E}_{P_\star(\cdot|x^h, a^h)} \sqrt{\frac{\nu dP_0(\cdot|x^h, a^h) + (1-\nu)d\tilde{P}(\cdot|x^h, a^h)}{dP_\star(\cdot|x^h, a^h)}}$$
$$\leq (\tilde{R}(x^h, a^h) - R_\star(x^h, a^h))^2 + 2 - 2 \mathbb{E}_{P_\star(\cdot|x^h, a^h)} \left[ \nu\sqrt{\frac{dP_0(\cdot|x^h, a^h)}{dP_\star(\cdot|x^h, a^h)}} + (1-\nu)\sqrt{\frac{d\tilde{P}(\cdot|x^h, a^h)}{dP_\star(\cdot|x^h, a^h)}} \right]$$

$$\leq \xi + \nu,$$

where the second inequality follows from Jensen's inequality applied to the function $\sqrt{\cdot}$. Now we can further invoke Sason and Verdú [2016, Theorem 9] to conclude that

$$\text{KL}(P_\star(\cdot|x^h, a^h)||P'(\cdot|x^h, a^h)) \leq \zeta(B/\nu)D_H(P'(\cdot|x^h, a^h), P_\star(\cdot|x^h, a^h))^2,$$

where $\zeta(b) = 2\log e$ if $b = 1$ and is equal to $\max\left\{ 1, \frac{b\log b + (1-b)\log e}{(1-\sqrt{b})^2} \right\}$ for all other $b > 0$. With this bound, we obtain that

$$\tilde{\ell}(M', x^h, a^h) = (\tilde{R}(x^h, a^h) - R_\star(x^h, a^h))^2 + \text{KL}(P_\star(\cdot|x^h, a^h)||P'(\cdot|x^h, a^h))$$
$$\leq \zeta\left(\frac{B}{\nu}\right)(\xi + \nu),$$

so that $M'$ is guaranteed to be in the class $\mathcal{M}(\epsilon)$ if $\xi \le \epsilon/(2\zeta(B/\nu))$ and $\nu \le \epsilon/(2\zeta(B/\nu))$. We focus on the case $B \ne \nu$, since $B = \nu \le \epsilon/(4\log e)$ is only of interest for very large values of $\epsilon$. For $B \ne \nu$, we first obtain an upper bound on $\zeta(B/\nu)$ in the regime where $\nu \le B/9$. That is, we upper bound $\zeta(b)$ for $b > 9$ so that $\sqrt{b} \le b/3$, where we have

$$\zeta(b) \le \frac{b \log b}{(1 - \sqrt{b})^2} \le \frac{b \log b}{b - 2\sqrt{b}} \le 3 \log b.$$

That is, when $B/\nu \ge 9$, it suffices to find a $\nu$ satisfying $\nu \le \epsilon/(6\log(B/\nu))$. Supposing that $B \ge \log(6B^2/\epsilon)$, a feasible setting is given by $\nu = \epsilon/(6\log(6B^2/\epsilon))$. With this setting of $\nu$, the stated condition on $\epsilon$ ensures that $B/\nu \ge 9$ as we required.

Similarly, we set $\xi = \epsilon/(6\log(B/\nu))$, which satisfies the desired bound on $\xi$. Finally, we can choose $p_0^\epsilon$ to be the uniform distribution on $\mathcal{C}_\nu(\xi)$ with the stated values of $\nu$ and $\xi$, and the upper bound follows from the same argument as in the first part.

# D   Details of examples for $Q$-type models

In this section we detail examples of concrete models which fall under our $Q$-type assumptions.

## D.1   Kernelized Non-linear Regulator

In the KNR model [Kakade et al., 2020],
$$x^{h+1} = W^\star \varphi(x^h, a^h) + \epsilon, \quad \text{with} \quad \epsilon \sim \mathcal{N}(0, \sigma^2 I),$$
for a known feature map $\varphi$, and the reward function is typically considered known. Since the reward is known, we consider discriminators $f(x, a, r, x') = g(x, a, x')$ and in particular choose the class $\mathcal{F} = \{f_v(x, a, r, x') = v^T x' : \|v\|_2 \le 1\}$. For this choice, we have for any model $M$:
$$(P_M f)(x^h, a^h) - (P^\star f)(x^h, a^h) = v^T(W_M - W^\star)\varphi(x^h, a^h),$$
for each $x^h, a^h$, where $W_M$ is the parameter underlying the model $M$. Furthermore, we have
$$\sup_{\|v\|_2 \le 1} \left| (P_M f)(x^h, a^h) - (P^\star f)(x^h, a^h) \right| = \|(W_M - W^\star)\varphi(x^h, a^h)\|_2$$
$$= \sigma\sqrt{2\mathrm{KL}(P_M(\cdot|x^h, a^h)\|P^\star(\cdot|x^h, a^h))}$$
$$\ge 2\sigma\mathrm{TV}(P_M(\cdot|x^h, a^h), P^\star(\cdot|x^h, a^h))$$
$$\ge \sigma\mathcal{E}_B(M, x^h, a^h).$$

Consequently, we see that the KNR model satisfies both inequalities in Assumption 5 with $\kappa = \sigma$.

We now describe some useful properties of the KNR model, regarding covering number of the parameter space, which will be important in proving Corollary 3 later.

**Lemma 11.** For the KNR model, suppose that the features satisfy $\sup_{h, x^h, a^h} \|\varphi(x^h, a^h)\|_2 \le B$ and $\|W^\star\|_2 \le R$. Then there is a distribution $p_0$ such that for any $T$, we have $\omega(3HT, p_0) = \mathcal{O}\left((d_\varphi d_\mathcal{X}) \log \frac{R\sqrt{\min(d_\varphi, d_\mathcal{X})}BHT}{\sigma}\right)$.

*Proof.* Let $\mathcal{M}_{KNR} = \{W \in \mathbb{R}^{d_\mathcal{X} \times d_\varphi} : \|W\|_2 \le R\}$ denote the model class for the KNR model. We prove the lemma by first constructing a cover of $\mathcal{M}_{KNR}$ such that at least one element in the cover lies in the set $\mathcal{M}_{KNR}(\epsilon)$, before demonstrating the bound by instantiating $p_0$ to be uniform over the cover.

We first begin with the value function in the initial state. Given any two models $M, M' \in \mathcal{M}$, we have

$$V_M(x^1) - V_{M'}(x^1) \le V_M(x^1) - V_{M'}^{\pi_M}(x^1) = \sum_{h=1}^{H} \mathbb{E}_{x^h, a^h \sim \pi_M}^{M'} \left[ Q_M^h(x^h, a^h) - (P_{M'}^h[r + V_M^{h+1}])(x^h, a^h) \right],$$

where the equality follows from the proof of Lemma 1. Since the reward function is known and shared across models, we can further simplify to get

$$V_M(x^1) - V_{M'}(x^1) \leq \sum_{h=1}^{H} \mathbb{E}_{x^h, a^h \sim \pi_M}^{M'} \left[ (P_M^h V_M^{h+1})(x^h, a^h) - (P_{M'}^h V_M^{h+1})(x^h, a^h) \right]$$

$$\leq 2 \sum_{h=1}^{H} \mathbb{E}_{x^h, a^h \sim \pi_M}^{M'} \mathrm{TV}(P_M^h(\cdot|x^h, a^h), P_{M'}^h(\cdot|x^h, a^h))$$

$$\leq \sum_{h=1}^{H} \mathbb{E}_{x^h, a^h \sim \pi_M}^{M'} \sqrt{\frac{1}{2} \mathrm{KL}(P_M^h(\cdot|x^h, a^h)||P_{M'}^h(\cdot|x^h, a^h))}$$

$$= \frac{1}{\sigma} \sum_{h=1}^{H} \mathbb{E}_{x^h, a^h \sim \pi_M}^{M'} \|(W_M - W^\star)\varphi(x^h, a^h)\|_2$$

$$\leq \frac{BH}{\sigma} \|W_M - W^\star\|_2,$$

where we use that $\|\varphi(x^h, a^h)\|_2 \leq B$ for all $h$ and $x^h, a^h$. A similar bound follows in the other direction, so that we get

$$\left| V_M(x^1) - V_{M'}(x^1) \right| \leq \frac{BH}{\sigma} \|W_M - W^\star\|_2. \tag{7}$$

For the KL divergence term, the argument is simpler and already implied by our earlier calculation, since we have

$$\sup_{h, x^h, a^h} \mathrm{KL}((P_M^h(\cdot|x^h, a^h)||P_{M'}^h(\cdot|x^h, a^h))) \leq \frac{B^2}{2\sigma^2} \|W_M - W^\star\|_2^2. \tag{8}$$

Now we note that picking any $\epsilon \leq 2H^2$, we are guaranteed that

$$\frac{BH}{\sigma} \|(W_M - W^\star)\|_2 \leq \epsilon \quad \Rightarrow \quad \frac{B^2}{2\sigma^2} \|(W_M - W^\star)\|_2^2 \leq \epsilon.$$

Hence, it suffices to cover the set $\mathcal{M}_{KNR}$ to a resolution of $\sigma\epsilon/BH$.

Let $R = \|W^\star\|_2$. Using Lemma 12, we see that the covering number of $\mathcal{M}_{KNR}$ to the desired resolution is at most $(CR\sqrt{\min(d_\varphi, d_\mathcal{X})}BH/(\sigma\epsilon))^{d_\varphi d_\mathcal{X}}$, for a universal constant $C$. Taking $p_0$ to be the uniform distribution over the elements of this cover, and observing that there exists some $M$ in the cover such that $M \in \mathcal{M}(\epsilon)$, we find that

$$\omega(3HT, p_0) = \inf_{\epsilon > 0} 3HT\epsilon + d_\varphi d_\mathcal{X} \log \frac{1}{\epsilon} + \mathcal{O}\left( d_\varphi d_\mathcal{X} \log \frac{R\sqrt{\min(d_\varphi, d_\mathcal{X})}BH}{\sigma} \right)$$

$$\leq 1 + d_\varphi d_\mathcal{X} \log(3HT) + \mathcal{O}\left( d_\varphi d_\mathcal{X} \log \frac{R\sqrt{\min(d_\varphi, d_\mathcal{X})}BH}{\sigma} \right),$$

which completes the proof. $\qquad\square$

**Lemma 12.** For any $d_1 \leq d_2$, let $\mathcal{S} = \{A \ : \ A \in \mathbb{R}^{d_1 \times d_2}, \|A\|_2 \leq R\}$. Then the $\epsilon$-covering number of $A$ is at most $(CR\sqrt{d_1}/\epsilon)^{d_1 d_2}$, for a universal constant $C$.

*Proof.* Let $A, B \in \mathbb{R}^{d_1 \times d_2}$ satisfy $\|A\|_2, \|B\|_2 \leq R$. Since $\|A\|_F/\sqrt{d_1} \leq \|A\|_2 \leq \|A\|_F \sqrt{d_1}$ when $d_1 \leq d_2$, we note that $\mathcal{S} \subseteq \{A \ : \ \|A\|_F \leq R\sqrt{d_1}\}$. Let $\mathrm{vec}(A) \in \mathbb{R}^{d_1 d_2}$ represent a vector formed by concatenating the columns of $A$, so that $\|A\|_F = \|\mathrm{vec}(A)\|_2$. Further observing that

$$\|A - B\|_2 \leq \|A - B\|_F = \|\mathrm{vec}(A) - \mathrm{vec}(B)\|_2,$$

we note that it suffices to choose a covering of the $\ell_2$ ball of radius $R\sqrt{d_1}$ in $\mathbb{R}^{d_1 d_2}$ in $\ell_2$-norm, to an accuracy $\epsilon$. Through the standard volumetric argument [Tomczak-Jaegermann, 1989, Vershynin, 2018], this covering number is at most $\mathcal{O}\left( \left(1 + \frac{2R\sqrt{d_1}}{\epsilon}\right)^{d_1 d_2} \right)$, which completes the proof. $\qquad\square$

## D.2 Linear Mixture MDP

In a linear mixture MDP [Modi et al., 2020, Ayoub et al., 2020, Zhou et al., 2021], we assume there exists a set of base models $\mathcal{M}_0 = \{M_1, \ldots, M_d\}$, and any model in $\mathcal{M}$ can be written as a mixture model of $\mathcal{M}_0$ as

$$M = \sum_{j=1}^{d} \nu_j M_j,$$

where $\nu_j \geq 0$ are unknown coefficients such that $\sum_j \nu_j = 1$. Here we consider the slightly more general form which is used in Du et al. [2021], who consider models of the form:

$$P_M(x^{h+1}|x^h, a^h) = \theta_M^\top \phi(x^h, a^h, x^{h+1}), \quad \text{and} \quad R_M(x^h, a^h) = \tilde{\theta}_M^\top \tilde{\phi}(x^h, a^h),$$

where $\phi$ and $\tilde{\phi}$ are known feature maps.

We now give a general factorization assumption which is fulfilled in linear mixture MDPs.

**Assumption 6** (Model dependent discriminator) *Let $f_M = r + g_M(x, a, x')$ be a function that depends on $M$. Then for any $x^h, a^h, h \in [H]$ and $M, M' \in \mathcal{M}$, there exist mappings $\psi^h(x^h, a^h, M')$ and $u^h(M)$, such that*

$$|(P_M f_{M'})(x^h, a^h) - (P^\star f_{M'})(x^h, a^h)| \geq |\langle \psi^h(x^h, a^h, M'), u^h(M)\rangle|, \quad \text{and}$$
$$|\langle \psi^h(x^h, a^h, M), u^h(M)\rangle| \geq \kappa \mathcal{E}_B(M, x^h, a^h). \tag{9}$$

*We assume that $\|u^h(M)\|_2 \leq B_1$.*

In a linear mixture MDP, we see that for $f_M(x, a, r, x') = r + V_M(x')$, we have for any pair $M, M' \in \mathcal{M}$:

$$(P_M f_{M'})(x^h, a^h) - (P^\star f_{M'})(x^h, a^h)$$
$$= (\tilde{\theta}_M - \tilde{\theta}_\star)^\top \tilde{\phi}(x^h, a^h) + (\theta_M - \theta^\star)^\top \int_x \phi(x^h, a^h, x) V_{M'}(x) dx.$$

Thus, we can define $\psi^h(x^h, a^h, M') = (\tilde{\phi}(x^h, a^h), \int_x \phi(x^h, a^h, x) V_{M'}(x) dx)$. Since using $f_M$ as the discriminator exactly recovers Bellman error (Lemma 1), the assumption holds with $\kappa = 1$ in this case.

## E Proofs of $V$-type decoupling results

In this section, we prove Propositions 3 and 4. We begin with two common lemmas that are required in both the proofs. The first lemma allows us to relate the IPM-style divergence measured in Assumption 3 with the mean squared error of rewards and Hellinger distance of dynamics to the ground truth on the RHS of Definition 2.

**Lemma 13.** Let $\mathcal{F}$ be a function class such that any $f \in \mathcal{F}$ satisfies $f(x, a, r, x') = r + g(x, a, x')$ for some $g \in \mathcal{G}$ with $g(x, a, x') \in [0, 1]$. Then we have for all $h, x^h, a^h$ and $M \in \mathcal{M}$:

$$\sup_{f \in \mathcal{F}} \left[ \left( P_M^h f(x^h, a^h) - P_\star^h f(x^h, a^h) \right)^2 \right]$$
$$\leq 4 \left[ D_H \left( P_M^h(\cdot \mid x^h, a^h), P_\star^h(\cdot \mid x^h, a^h) \right)^2 + \left( R_M^h(x^h, a^h) - R_\star^h(x^h, a^h) \right)^2 \right],$$

where

$$D_H(P, P') = \mathbb{E}_{x \sim P} (\sqrt{dP'(x)/dP(x)} - 1)^2.$$

*Proof.* We have

$$\sup_{f \in \mathcal{F}} \left[ \left( P_M^h f(x^h, a^h) - P_\star^h f(x^h, a^h) \right)^2 \right]$$

$$= \sup_{g \in \mathcal{G}} \left[ \left( P_M^h g(x^h, a^h) + R_M^h(x^h, a^h) - P_\star^h g(x^h, a^h) - R_\star^h(x^h, a^h) \right)^2 \right]$$

$$\leq 2 \sup_{g \in \mathcal{G}} \left[ \left( P_M^h g(x^h, a^h) - P_\star^h g(x^h, a^h) \right)^2 + \left( R_M^h(x^h, a^h) - R_\star^h(x^h, a^h) \right)^2 \right]$$

$$\leq 2 \left[ 2\mathrm{TV} \left( P_M^h(\cdot \mid x^h, a^h), P_\star^h(\cdot \mid x^h, a^h) \right)^2 + \left( R_M^h(x^h, a^h) - R_\star^h(x^h, a^h) \right)^2 \right]$$

$$\leq 4 \left[ D_H \left( P_M^h(\cdot \mid x^h, a^h), P_\star^h(\cdot \mid x^h, a^h) \right)^2 + \left( R_M^h(x^h, a^h) - R_\star^h(x^h, a^h) \right)^2 \right],$$

where $\mathrm{TV}(P, P') = \frac{1}{2} \mathbb{E}_{x \sim P} |dP'(x)/dP(x) - 1|$. $\qquad\square$

The next lemma allows us to decouple the Bellman error on the LHS of Definition 2 in the distribution over states.

**Lemma 14** (Witness rank decoupling over states). Let $\mathcal{F}$ be a function class satisfying Assumption 3. Then for any $\mu_1, \epsilon_1 > 0$, context $x^1$ and distribution $p \in \Delta(\mathcal{M})$, we have

$$\kappa \mathbb{E}_{M \sim p} \mathcal{E}_B(M, M, h, x^1) \leq \frac{\mu}{2} \mathbb{E}_{\substack{M \sim p \\ M' \sim p}} \sup_{f \in \mathcal{F}} \mathbb{E}_{x^h \sim \pi_M | x^1} \mathbb{E}_{a^h \sim \pi_{M'}(\cdot | x^h)} \left[ \left( P_{M'}^h f(x^h, a^h) - P_\star^h f(x^h, a^h) \right)^2 \right]$$

$$+ \frac{1}{2\mu} d_{\mathrm{eff}}(\psi^h, \epsilon) + B_1 \epsilon.$$

*Proof.* We need some additional notation for the proof. Given a distribution $p \in \Delta(\mathcal{M})$, we define:

$$\Sigma^h(p, x^1) = \mathbb{E}_{M \sim p} \psi^h(M, x^1) \otimes \psi^h(M, x^1),$$

$$K^h(\mathcal{F}, \lambda) = \sup_{p, x^1} \mathrm{trace}((\Sigma^h(p, x^1) + \lambda I)^{-1} \Sigma^h(p, x^1)).$$

We have for any $\lambda > 0$

$$\kappa \mathbb{E}_{M \sim p} \mathcal{E}_B(M, M, h, x^1) \leq \mathbb{E}_{M \sim p} \left| \left\langle \psi^h(M, x^1), u^h(M, x^1) \right\rangle \right| \qquad \text{(Assumption 3)}$$

$$\leq \sqrt{\mathbb{E}_{M \sim p} \| u^h(M, x^1) \|_{\Sigma^h(p, x^1) + \lambda I}^2 \; \mathbb{E}_{M \sim p} \| \psi^h(M, x^1) \|_{(\Sigma^h(p, x^1) + \lambda I)^{-1}}^2}$$

$$\leq \sqrt{K^h(\mathcal{F}, \lambda) \left( \mathbb{E}_{M \sim p} \| u^h(M, x^1) \|_{\Sigma^h(p, x^1)}^2 + \lambda \mathbb{E}_{M \sim p} \| u^h(M, f, x^1) \|^2 \right)}$$

$$\leq \sqrt{K^h(\mathcal{F}, \lambda) \mathbb{E}_{M \sim p} \| u^h(M, x^1) \|_{\Sigma^h(p, x^1)}^2} + B_1 \sqrt{\lambda K^h(\mathcal{F}, \lambda)}$$

$$\leq \frac{\mu_1}{2} \mathbb{E}_{M \sim p, M' \sim p} \left\langle u^h(M, x^1), \psi^h(M', x^1) \right\rangle^2 + \frac{1}{2\mu_1} K^h(\mathcal{F}, \lambda) + B_1 \sqrt{\lambda K^h(\mathcal{F}, \lambda)}.$$

Further optimizing over the choice of $\lambda > 0$ such that $\lambda K^{h-1}(\lambda) \leq \epsilon^2$, we see that

$$\kappa \mathbb{E}_{M \sim p} \mathcal{E}_B(M, M, h, x^1)$$

$$\leq \frac{\mu_1}{2} \underbrace{\mathbb{E}_{M \sim p, M' \sim p} \left\langle u^h(M, x^1), \psi^h(M', x^1) \right\rangle^2}_{\mathcal{T}_1} + \frac{1}{2\mu_1} d_{\mathrm{eff}}(\psi^h, \epsilon) + B_1 \epsilon. \qquad (10)$$

Now we focus on $\mathcal{T}_1$. In fact, it follows by Assumption 3 that

$$\mathcal{T}_1 \leq \mathop{\mathbb{E}}_{M \sim p, M' \sim p} \left[ \left( \sup_{f \in \mathcal{F}} \mathop{\mathbb{E}}_{x^h \sim \pi_M | x^1} \mathop{\mathbb{E}}_{a^h \sim \pi_{M'}(\cdot | x^h) | x^1} \left| P^h_{M'} f(x^h, a^h) - P^h_\star f(x^h, a^h) \right| \right)^2 \right]$$

$$\leq \mathop{\mathbb{E}}_{M \sim p, M' \sim p} \sup_{f \in \mathcal{F}} \mathop{\mathbb{E}}_{x^h \sim \pi_M | x^1} \mathop{\mathbb{E}}_{a^h \sim \pi_{M'}(\cdot | x^h) | x^1} \left[ \left( P^h_{M'} f(x^h, a^h) - P^h_\star f(x^h, a^h) \right)^2 \right], \qquad (11)$$

which implies the desired bound. $\qquad \square$

### E.1 Decoupling for finite actions and known embedding features

Given the above lemmas, the proof of Proposition 3 is immediate by changing the distribution over $a^h$ to the uniform distribution via importance sampling. In fact, here we state and prove a slight generalization of the proposition which applies to settings where the features $\phi^h$ in Assumption 4 are known, of which the finite action setting is a special case. For notational ease, we define the $G$-optimal design in a feature map $\phi$ as the policy which optimizes:

$$\pi_{des}(\cdot | x, \phi) = \mathop{\arginf}_{\pi \in \Delta(\mathcal{A})} \sup_{a \in \mathcal{A}} \phi(x, a)^\top \left( \mathop{\mathbb{E}}_{a' \sim \pi} \phi(x, a') \phi(x, a')^\top \right)^{-1} \phi(x, a). \qquad (12)$$

If the inverse does not exist for some $x$, we can add a small $\epsilon I$ offset to make the covariance invertible, and all of our subsequent conclusions hold in the limit of $\epsilon$ decreasing to 0. We omit this technical development in the interest of conciseness. With slight abuse of notation, we also use $\pi^\phi_{des}$ to be the policy such that $\pi^\phi_{des}(a|x) = \pi_{des}(a|x, \phi)$. It is evident from the choice of $\phi^h(x, a) = e_a$ for finite action problems that $\pi^\phi_{des} = \mathrm{Unif}(\mathcal{A})$ in this case.

**Proposition 6** (Design-based decoupling) *Under Assumptions 3 and 4, suppose further that the map $\phi^h$ is known with $\dim(\phi^h) = d(\phi^h) < \infty$. Let the effective dimensions of $\psi^h$ and $\phi^h$ be as instantiated in Proposition 4. Then for any $\epsilon > 0$, we have*

$$\mathrm{dc}(\epsilon, p, p \circ^h \pi^{\phi^h}_{des}, 0.5) \leq \frac{4d(\phi^h)}{\kappa^2} d_{\mathrm{eff}} \left( \psi^h, \frac{\kappa}{B_1} \epsilon \right).$$

*Proof.* Invoking Lemma 14, we have

$$\kappa \mathop{\mathbb{E}}_{M \sim p} \mathcal{E}_B(M, M, h, x^1) \leq \frac{\mu}{2} \mathop{\mathbb{E}}_{\substack{M \sim p \\ M' \sim p}} \sup_{f \in \mathcal{F}} \mathop{\mathbb{E}}_{x^h \sim \pi_M | x^1} \mathop{\mathbb{E}}_{a^h \sim \pi_{M'}(\cdot | x^h)} \left[ \left( P^h_{M'} f(x^h, a^h) - P^h_\star f(x^h, a^h) \right)^2 \right]$$

$$+ \frac{1}{2\mu} d_{\mathrm{eff}}(\psi^h, \epsilon) + B_1 \epsilon.$$

Let $\Sigma^h(x) = \mathbb{E}_{a' \sim \pi_{des}(\cdot | x, \phi^h)} \phi^h(x, a') \phi^h(x, a')^\top$. Now invoking Assumption 4, we further observe for any $x^h, a^h$ and $f \in \mathcal{F}$:

$$\left( P^h_{M'} f(x^h, a^h) - P^h_\star f(x^h, a^h) \right)^2$$
$$= \left\langle w^h(M, f, x^h), \phi^h(x^h, a^h) \right\rangle^2$$
$$\leq \| w^h(M, f, x^h) \|^2_{\Sigma^h(x^h)} \| \phi^h(x^h, a^h) \|^2_{\Sigma^h(x^h)^{-1}} \qquad \text{(Cauchy-Schwarz)}$$
$$\leq \sup_a \| \phi^h(x^h, a) \|^2_{\Sigma^h(x^h)^{-1}} \mathop{\mathbb{E}}_{a \sim \pi_{des}(\cdot | x^h, \phi^h)} \left\langle w^h(M, f, x^h), \phi^h(x^h, a^h) \right\rangle^2$$
$$\leq d(\phi^h) \mathop{\mathbb{E}}_{a \sim \pi_{des}(\cdot | x^h, \phi^h)} \left( P^h_{M'} f(x^h, a^h) - P^h_\star f(x^h, a^h) \right)^2,$$

where the last inequality is due to the Kiefer-Wolfowitz theorem [Kiefer and Wolfowitz, 1960]. Hence, we have shown that

$$
\mathop{\mathbb{E}}_{M \sim p} \mathcal{E}_B(M, M, h, x^1)
$$

$$
\leq \frac{\mu d(\phi^h)}{2\kappa} \mathop{\mathbb{E}}_{\substack{M \sim p \\ M' \sim p}} \sup_{f \in \mathcal{F}} \mathop{\mathbb{E}}_{x^h \sim \pi_M | x^1} \mathop{\mathbb{E}}_{a^h \sim \pi_{des}(\cdot | x^h, \phi^h)} \left[ \left( P_{M'}^h f(x^h, a^h) - P_\star^h f(x^h, a^h) \right)^2 \right]
$$

$$
+ \frac{1}{2\mu\kappa} d_{\text{eff}}(\psi^h, \epsilon) + \frac{B_1 \epsilon}{\kappa}
$$

$$
\leq \sqrt{ \frac{d(\phi^h) d_{\text{eff}}(\psi^h, \epsilon)}{\kappa^2} \mathop{\mathbb{E}}_{\substack{M \sim p \\ M' \sim p}} \sup_{f \in \mathcal{F}} \mathop{\mathbb{E}}_{x^h \sim \pi_M | x^1} \mathop{\mathbb{E}}_{a^h \sim \pi_{des}(\cdot | x^h, \phi^h)} \left( P_{M'}^h f(x^h, a^h) - P_\star^h f(x^h, a^h) \right)^2 } + \frac{B_1 \epsilon}{\kappa}.
$$

Further applying Lemma 13 and redefining $\epsilon \to B_1 \epsilon / \kappa$ gives the result. $\qquad \square$

## E.2 Proof of Proposition 4

Now we prove Proposition 4, which requires a more careful decoupling argument over the actions, since the features $\phi^h$ are not known. In this case, we need the following decoupling lemma over the actions, given an observation, which can then be combined with Lemma 14.

Now we focus on a fixed observation $x^h$ at some level $h$. Given any measure $p \in \Delta(\mathcal{M})$, let us define

$$
\widetilde{\Sigma}^h(p, x^h) = \mathop{\mathbb{E}}_{M \sim p, a \sim \pi_M(\cdot | x^h)} \phi^h(x^h, a) \otimes \phi^h(x^h, a),
$$

$$
\tilde{K}^h(\mathcal{F}, \lambda) = \sup_{p, x} \text{trace}((\widetilde{\Sigma}^h(p, x^h) + \lambda I)^{-1} \widetilde{\Sigma}^h(p, x^h)).
$$

We have the following decoupling result for the choice of actions, given a context.

**Lemma 15** (Linear embedding decoupling)**.** Let $\mathcal{F}$ be any function class satisfying Assumption 4. Then we have for any model $M \in \mathcal{M}$, level $h$, observation $x^h$ and constants $\mu_2, \epsilon_2 > 0$

$$
\mathop{\mathbb{E}}_{a^h \sim \pi_M(\cdot | x^h)} \left[ \left( P_M^h f(x^h, a^h) - P_\star^h f(x^h, a^h) \right)^2 \right]
$$

$$
\leq \mu_2 \mathop{\mathbb{E}}_{M' \sim p} \mathop{\mathbb{E}}_{a^h \sim \pi_{M'}(\cdot | x^h)} \left( P_M^h f(x^h, a^h) - P_\star^h f(x^h, a^h) \right)^2 + \frac{1}{\mu_2} d_{\text{eff}}^h(\phi^h, \epsilon_2) + B_2 \epsilon_2.
$$

*Proof.* Under Assumption 4, we have

$$
\mathop{\mathbb{E}}_{a^h \sim \pi_M(\cdot | x^h)} \left[ \left( P_M^h f(x^h, a^h) - P_\star^h f(x^h, a^h) \right)^2 \right]
$$

$$
\leq 2 \mathop{\mathbb{E}}_{a^h \sim \pi_M(\cdot | x^h)} \left[ \left| P_M^h f(x^h, a^h) - P_\star^h f(x^h, a^h) \right| \right] \qquad \text{(Since } f \in [0, 2])
$$

$$
= 2 \mathop{\mathbb{E}}_{a^h \sim \pi_M(\cdot | x^h)} \left| \left\langle w^h(M, f, x^h), \phi^h(x^h, a^h) \right\rangle \right| \qquad \text{(Assumption 4)}
$$

$$
\leq 2 \sqrt{ \mathop{\mathbb{E}}_{a^h \sim \pi_M(\cdot | x^h)} \| w^h(M, f, x^h) \|_{\widetilde{\Sigma}(p, x^h) + \lambda I}^2 \tilde{K}^h(\mathcal{F}, \lambda) }
$$

$$
\leq 2 \sqrt{ \left( \mathop{\mathbb{E}}_{M' \sim p} \mathop{\mathbb{E}}_{a^h \sim \pi_{M'}(\cdot | x^h)} \left( P_M^h f(x^h, a^h) - P_\star^h f(x^h, a^h) \right)^2 + \lambda B_2^2 \right) \tilde{K}^h(\mathcal{F}, \lambda) }
$$

$$
\leq \mu_2 \mathop{\mathbb{E}}_{M' \sim p} \mathop{\mathbb{E}}_{a^h \sim \pi_{M'}(\cdot | x^h)} \left( P_M^h f(x^h, a^h) - P_\star^h f(x^h, a^h) \right)^2 + \frac{1}{\mu_2} \tilde{K}^h(\mathcal{F}, \lambda) + 2 B_2 \sqrt{\lambda \tilde{K}^h(\mathcal{F}, \lambda)}.
$$

Further optimizing over the choice of $\lambda$ yields the bound for any $x^h$

$$\mathop{\mathbb{E}}_{a^h \sim \pi_M(\cdot | x^h)} \left[ \left( P_M^h f(x^h, a^h) - P_\star^h f(x^h, a^h) \right)^2 \right]$$

$$\leq \mu_2 \mathop{\mathbb{E}}_{M' \sim p} \mathop{\mathbb{E}}_{a^h \sim \pi_{M'}(\cdot | x^h)} \left( P_M^h f(x^h, a^h) - P_\star^h f(x^h, a^h) \right)^2 + \frac{1}{\mu_2} d_{\text{eff}}^h(\phi^h, \epsilon_2) + B_2 \epsilon_2. \qquad (13)$$

This leads to the desired bound. $\qquad \square$

With these lemmas, we can now prove Proposition 4.

*Proof of Proposition 4.* Hence, we have for any level $h$ and context $x^1$:

$$\kappa \mathop{\mathbb{E}}_{M \sim p} \mathcal{E}_B(M, M, h, x^1) = \kappa \mathop{\mathbb{E}}_{M \sim p} \mathop{\mathbb{E}}_{x^h, a^h \sim \pi_M(\cdot | x^1)} \mathcal{E}_B(M, x^h, a^h)$$

$$\leq \frac{\mu_1}{2} \mathop{\mathbb{E}}_{M, M' \sim p} \sup_{f \in \mathcal{F}} \mathop{\mathbb{E}}_{x^h \sim \pi_M | x^1} \mathop{\mathbb{E}}_{a^h \sim \pi_{M'}(\cdot | x^h)} \left[ \left( P_M^h f(x^h, a^h) - P_\star^h f(x^h, a^h) \right)^2 \right] + \frac{1}{2\mu_1} d_{\text{eff}}(\psi^h, \epsilon_1) + B_1 \epsilon_1$$

$$\text{(Lemma 14)}$$

$$\leq \frac{\mu_1 \mu_2}{2} \mathop{\mathbb{E}}_{M, M', M'' \sim p} \sup_{f \in \mathcal{F}} \mathop{\mathbb{E}}_{x^h \sim \pi_{M''} | x^1} \mathop{\mathbb{E}}_{a^h \sim \pi_{M'}(\cdot | x^h)} \left[ \left( P_M^h f(x^h, a^h) - P_\star^h f(x^h, a^h) \right)^2 \right]$$

$$+ \frac{1}{2\mu_1} d_{\text{eff}}(\psi^h, \epsilon_1) + B_1 \epsilon_1 + \frac{\mu_1}{2\mu_2} d_{\text{eff}}(\phi^h, \epsilon_2) + \frac{\mu_1 B_2 \epsilon_2}{2}. \qquad \text{(Lemma 15)}$$

Now we optimize over the choices of $\mu_1, \mu_2$ and plug in Lemma 13 as before to complete the proof. $\qquad \square$

# F Proofs of $Q$-type decoupling results

In this section, we prove Proposition 5, as well as the following result for decoupling under Assumption 6.

**Proposition 7** *Under Assumption 6, let $\chi = \psi^h(x^h, a^h, M')$ with $z_2 = (x^h, a^h, M')$ in Definition 1. Then for any $\epsilon > 0$ we have*

$$\text{dc}(\epsilon, p, \pi_{\text{gen}}(h, p), 0.5) \leq \frac{4}{\kappa^2} d_{\text{eff}}(\psi^h, (\kappa/B_1)\epsilon),$$

*where $\pi_{\text{gen}}(h, p) = \pi_M$ with $M \sim p$.*

## F.1 Proof of Proposition 5

*Proof.* Let us define for any distribution $p \in \Delta(\mathcal{M})$ and $x$:

$$\Sigma_Q^h(p, x) = \mathop{\mathbb{E}}_{M \sim p} \mathop{\mathbb{E}}_{a \sim \pi_M(\cdot | x)} \psi^h(x, a) \otimes \psi^h(x, a),$$

$$K_Q^h(\mathcal{F}, \lambda) = \sup_{p, x} \text{trace}((\Sigma_Q^h(p, x) + \lambda I)^{-1} \Sigma_Q^h(p, x)).$$

We have for any $\lambda > 0$, $x^1$ and $h \in [H]$

$$\kappa \underset{M\sim p}{\mathbb{E}} \underset{x^h,a^h\sim\pi_M|x^1}{\mathbb{E}} \mathcal{E}_B(M,h,x^h,a^h) \le \underset{M\sim p}{\mathbb{E}} \underset{x^h,a^h\sim\pi_M}{\mathbb{E}} \sup_{f\in\mathcal{F}} \left|\langle \psi^h(x^h,a^h), u^h(M,f)\rangle\right|$$

(Assumption 5)

$$\le \underset{M\sim p}{\mathbb{E}} \underset{x^h,a^h\sim\pi_M|x^1}{\mathbb{E}} \sqrt{\sup_{f\in\mathcal{F}}\|\psi(x^h,a^h)\|^2_{(\Sigma_Q^h(x^h,p)+\lambda I)^{-1}}\|u^h(M,f)\|^2_{\Sigma_Q^h(x^h,p)+\lambda I}}$$

$$\le \sqrt{\underset{M\sim p}{\mathbb{E}} \underset{x^h,a^h\sim\pi_M|x^1}{\mathbb{E}} \|\psi(x^h,a^h)\|^2_{(\Sigma_Q^h(x^h,p)+\lambda I)^{-1}}\left(\sup_{f\in\mathcal{F}}\|u^h(M,f)\|^2_{\Sigma_Q^h(x^h,p)}+\lambda B_1^2\right)} \quad \text{(Jensen's)}$$

$$\le B_1\sqrt{\lambda K_Q^h(\lambda)} + \frac{\mu}{2}\underset{M\sim p}{\mathbb{E}}\sup_{f\in\mathcal{F}}\underset{x^h,a^h\sim\pi_M|x^1}{\mathbb{E}}\|u^h(M,f)\|^2_{\Sigma_Q^h(x^h,p)} + \frac{1}{2\mu}K_Q^h(\lambda)$$

$$\le \frac{\mu}{2}\underset{M,M'\sim p}{\mathbb{E}} \underset{x^h,a^h\sim\pi_M|x^1}{\mathbb{E}}\sup_{f\in\mathcal{F}}\left(P_{M'}^h f(x^h,a^h)-P_\star^h(x^h,a^h)\right)^2 + \frac{1}{2\mu}K_Q^h(\lambda) + B_1\sqrt{\lambda K_Q^h(\lambda)}$$

$$\le \frac{\mu}{2}\underset{M,M'\sim p}{\mathbb{E}} \underset{x^h,a^h\sim\pi_M|x^1}{\mathbb{E}}\sup_{f\in\mathcal{F}}\left(P_{M'}^h f(x^h,a^h)-P_\star^h(x^h,a^h)\right)^2 + \frac{1}{2\mu}d_{\text{eff}}(\phi^h,\epsilon) + B_1\epsilon,$$

where the last line optimizes over the choice of $\lambda > 0$ such that $\lambda K_Q^h(\lambda) \le \epsilon^2$, which implies the desired bound. $\qquad\square$

### F.2   Proof of Proposition 7

*Proof.* Let us recall Assumption 6, and define

$$\Sigma^h(p,x^1) = \underset{M\sim p}{\mathbb{E}} \underset{x,a\sim\pi_M|x^1}{\mathbb{E}} \psi^h(x,a,M)\otimes\psi^h(x,a,M),$$

$$K^h(\lambda) = \sup_{p,x^1}\text{trace}((\Sigma^h(p,x^1)+\lambda I)^{-1}\Sigma^h(p,x^1)).$$

We have

$$\underset{M\sim p}{\mathbb{E}} \underset{(x^h,a^h)\sim\pi_M|x^1}{\mathbb{E}} \kappa\mathcal{E}_B(M,x^h,a^h)$$

$$\le \underset{M\sim p}{\mathbb{E}} \underset{(x^h,a^h)\sim\pi_M|x^1}{\mathbb{E}} \left|\langle\psi^h(x^h,a^h,M),u^h(M)\rangle\right|$$

$$\le \sqrt{\underset{M\sim p}{\mathbb{E}} u^h(M)^\top(\Sigma^h(p,x^1)+\lambda I)u^h(M)K^h(\lambda)}$$

$$\le \sqrt{K^h(\lambda)\underset{M'\sim p}{\mathbb{E}}\underset{M\sim p}{\mathbb{E}}\underset{(x^h,a^h)\sim\pi_{M'}|x^1}{\mathbb{E}}\left|\langle\psi^h(x^h,a^h,M'),u^h(M)\rangle\right|^2} + \sqrt{\lambda K^h(\lambda)}B_1$$

$$\le \sqrt{K^h(\lambda)\underset{M\sim p}{\mathbb{E}}\underset{M'\sim p}{\mathbb{E}}\underset{(x^h,a^h)\sim\pi_{M'}|x^1}{\mathbb{E}}\left|P_M f_{M'}(x^h,a^h)-P^\star f_{M'}(x^h,a^h)\right|^2} + \sqrt{\lambda K^h(\lambda)}B_1$$

$$\le \sqrt{4K^h(\lambda)\underset{M\sim p}{\mathbb{E}}\underset{M'\sim p}{\mathbb{E}}\underset{(x^h,a^h)\sim\pi_{M'}|x^1}{\mathbb{E}}\ell^h(M,x^h,a^h)} + \sqrt{\lambda K^h(\lambda)}B_1.$$

The last inequality used Lemma 13. $\qquad\square$