# OpenReview forum: "Model-based RL with Optimistic Posterior Sampling: Structural Conditions and Sample Complexity"
_NeurIPS.cc/2022/Conference — NeurIPS 2022 Accept_

### Official Review · Reviewer_sfzn · 2022-07-11

**Rating:** 6
**Confidence:** 3
**Soundness:** 3 good
**Presentation:** 3 good
**Contribution:** 3 good

**Summary:**

The authors propose a general algorithm for model-based RL based on optimistic posterior sampling. Theoretical analysis is provided to show that it achieves state-of-the-art sample complexity under some structural assumptions.

**Questions:**

It's related with the weakness discussed above. Can the authors provide further discussions on how we can reasonably quantify (or even estimate) these quantities in order to better understand the sample complexity in some general settings.

**Ethics Review Area:**

["I don’t know"]

**Limitations:**

Yes

**Strengths And Weaknesses:**

Strength: The proposed algorithm is an interesting effort to solve the problems in model-based RL that collects data from the environment given the current learning state and defines the quality of a model given the current acquired dataset. The idea is to use the data likelihood as the estimate for model quality, and use an optimistic posterior sampling method to collect data. A so-called Bellman error decoupling method is used for the sample complexity of near-optimal policy, and it is sample efficient under the condition that decoupling coefficient of a model-based RL is small.

Weakness: The assumptions in the sample complexity is somewhat stringent for the Hellinger decoupling. It's not clear under the general settings how to measure the Hellinger distance and Bellman Error to quantify such condition. It help better understand such condition if the authors can provide further discussions on how we can reasonably quantify (or even estimate) these quantities in order to better understand the sample complexity.

---

> ### Author Response · Authors · 2022-07-29
> **Thank you for the feedback. Please find responses to your questions below.**
>
> We thank you for your encouraging and constructive feedback, but we are perhaps somewhat puzzled by your review. Please note that our algorithm never needs to estimate a Bellman error or Hellinger suboptimality. Definition 2 is purely used in our analysis, and none of the quantities in it need to be directly estimated from data.
>
> Regarding examples of when Definition 2 is satisfied, Section 6 gives fairly broad examples of structural conditions under which Definition 2 holds with a reasonable bound on the decoupling coefficient. Some of the structural conditions in Section 6, such as V-type factorization which subsumes MDPs with low-rank dynamics and Q-type factorization which includes linear MDPs, linear mixture MDPs and several models in continuous control, cover and extend almost all standard models studied in theoretical RL.
>
> Concretely, Propositions 3 and 4 give explicit bounds on the decoupling coefficient in terms of other problem parameters for the V-type scenario, and Proposition 5 gives a bound for the Q-type seting. Further observe that in these examples, it is not necessary to separately estimate the Bellman error and Hellinger distance to establish the definition, but instead the structural assumptions allow us to bound one quantity in terms of the other.
>
> Perhaps we are missing the nature of examples that the reviewer has in mind, and we would be grateful if you would further explain this.
>
> We are certainly quite happy to further expand on the examples in the final version to incorporate the reviewer's feedback. Thanks again for the helpful comments.

---

> > ### Comment · Reviewer_sfzn · 2022-08-10
> > **Reply to authors response**
> >
> > Thanks for the response, which is helpful in clarifying some of my confusions in the initial review. Given that, I will adjust my reviews correspondingly.

---

### Official Review · Reviewer_93tk · 2022-07-11

**Rating:** 7
**Confidence:** 2
**Soundness:** 4 excellent
**Presentation:** 3 good
**Contribution:** 4 excellent

**Summary:**

The work proposed a unified framework for model-based RL by designing likelihood based posterior sampling methods. The theoretical framework unified many previously studied example scenarios and give state-of-art sample complexity in some cases. The authors also claim by combining variance reduction techniques, the sample complexity can be further improved.

**Questions:**

1. Extra T in Eq. (3)?

**Limitations:**

Yes.

**Strengths And Weaknesses:**

The paper discusses a meaningful unified approach for model-based RL. The algorithm and results are well presented. The theoretical analysis looks solid.

---

> ### Author Response · Authors · 2022-07-29
> **Thank you for the feedback. Please find responses to your questions below.**
>
> We thank you for your encouraging and constructive feedback. Thank you for spotting the typo in equation 3, there is indeed an extra $T$ factor. We will address this in the final version.

---

### Official Review · Reviewer_ysuF · 2022-07-11

**Rating:** 7
**Confidence:** 4
**Soundness:** 3 good
**Presentation:** 3 good
**Contribution:** 3 good

**Summary:**

The paper develops and formulates a statistical framework for model-based reinforcement learning with an optimistic posterior sampling mechanism. The paper develops a theoretical framework to solve the exploration-exploitation balance in an MBRL framework by incorporating an exploit-or-learn behavior in the learning framework- typical of optimistic posterior sampling algorithms. The paper draws an interesting connection between the regret of the optimistic PSRL algorithm with Hellinger distance and thereby derives a unified posterior sampling algorithm with state-of-the-art sample complexity under several structural assumptions.

**Questions:**

1. How does the bound in Definition 3 holds even under misspecified prior, can you please discuss that clearly?

2. How $\eta, \eta'$ in 4 in Algorithm 1 is obtained, and how much impact will it have on the analysis?

3. In [1], they show that the Bellman error is a poor proxy for the accuracy of the value function and the magnitude of the Bellman error is weakly related to the distance to the true value function. However, here the analysis has a very strong dependence on the Bellman error. Although it's a general issue and not specific to this research, please can you share your thoughts on the same.


**Limitations:**

The authors have mentioned the limitations of the current theoretical analysis including the need to perform empirical evidence, extension to non-linear function approximations in the Conclusion.

**Strengths And Weaknesses:**

The primary strength lies in the statistical formulation of an optimistic model-based posterior sampling reinforcement algorithm and the connection of the regret with the Hellinger distance between the conditional probabilities to derive the state-of-the-art sample complexity for MBRL under structural assumptions. As done in optimistic algorithms, the algorithm assigns an optimistic bonus as done by $p_t(M) \propto p_0(M)\exp(\sum_{s=1}^{t-1}(\gamma V_M(x_s^1) + L_s^{h_s}(M))$ which means that the posterior would concentrate on model which predicts the history well or models under which the policy will lead to high value function. The novelty lies in the decoupling framework which builds the connection between the Bellman error and the Hellinger distance $D_h (P_M(\cdot | x^h, a^h), P_\star(\cdot | x^h, a^h))$, with a dependence on the true dynamics. As it can be observed that it has a dependence on the true dynamics, hence it is critical to understand the confidence set creation to measure how
well the prior distribution covers the optimal model $M^*$ which is shown in Definition 3. Although, it needs a little more explicit explanation on how $M(\epsilon)$ is non-empty for $\epsilon > 0$ and also the bound still holds for misspecified prior. Overall the paper proposes a very interesting theoretical framework leveraging a Hellinger decoupling framework to derive the state-of-the-art sample complexity for optimistic PSRL-type algorithms and is also simplistic and unique in its formulation. The paper is well-written and the mathematical analysis is clear and precise. The main strength lies in the decoupling framework which is novel and very interesting and helps in controlling the Bellman error optimally. However, as pointed out by the authors it will be interesting to see the empirical evidence of the algorithm especially given that the theory has a strong dependence on the Bellman error and recent literature has shown that the Bellman error is a poor proxy for the accuracy of the value function [1].

References :
[1]. Why Should I Trust You, Bellman? The Bellman Error is a Poor Replacement for Value Error by Scott Fujimoto, David Meger, Doina Precup, Ofir Nachum, Shixiang Shane Gu. https://arxiv.org/abs/2201.12417

---

> ### Author Response · Authors · 2022-07-29
> **Thank you for the feedback. Please find responses to your questions below.**
>
> We thank you for your encouraging and constructive feedback. You raise an interesting point regarding the Bellman error, but note that the Bellman error is never used in our algorithm, but just the analysis. While the debates about good proxies for value errors, whether it is Bellman residuals, TD estimates or other alternatives are as old as the field itself at this point, we feel that this issue is somewhat tangential to our paper given the use of Bellman error as purely an analytical tool in the approximately well-specified case. Nevertheless, we will make a note of this in the final version of the paper, and leave other possibilities that might be more robust to misspecification to future work.
>
> Regarding the questions:
>
> 1. Please note that their is no issue of misspecified priors here. We are not analyzing the Bayesian regret that relies on a well-specified prior, and use, for instance the uniform prior for finite model classes. Perhaps the reviewer is asking about our comment following Definition 3 concerning misspecified model classes, where the model error cannot be made arbitrarily small. In such classes, there is an approximation error $\epsilon_0$ such that $p_0(\mathcal{M}(\epsilon) = 0$ for any $\epsilon < \epsilon_0$. Since this makes the second term in the definition of $\omega(\alpha, p_0)$ to be infinite, the minimization over $\epsilon$ never selects any $\epsilon < \epsilon_0$. We will expand on this point in the final version.
>
> 2. $\eta$ and $\eta'$ are set to 1/6 as per the theory in Proposition 2. Of course it is likely preferable to treat them as hyperparameters in practice and optimize them based on the quality of the learned model.
>
> 3. As discussed above, the Bellman error is never used in our algorithm. For our purposes, we rely on the appealing property of Bellman error to capture the model suboptimality via Lemma 1. We note that some of the issues noted in [1] are generally more pronounced in grossly misspecified settings, where error cancellation etc. might cause problems. A second issue they raise is incomplete datasets in offline RL, which we never encounter in the online exploration setting. We primarily focus on approximately well-specified settings here with online exploration, and investigating these issues in more detail for misspecified/offline settings can be an interesting direction for follow-up research.
>
> We thank the reviewer again for their helpful comments and will update the final draft to incorporate their comments and questions with appropriate discussion.

---

> > ### Comment · Reviewer_ysuF · 2022-08-07
> > **Confusion on Point 1**
> >
> > Thanks for the comments and it enhance my understanding of the paper.
> >
> > I absolutely agree with the author's point on the Bellman error issue and I agree that it was mainly for the analysis and not the algorithm.
> >
> > On Point 1, I specifically wanted to understand the aspects of Definition 3 for misspecified model classes. Although Point 1 explains the concern, still I am not absolutely clear on how it can work under any misspecified model class and would request the authors to provide a more detailed description. Thanks.

---

> > > ### Author Response · Authors · 2022-08-07
> > > **Thank you for the response**
> > >
> > > Thanks for reading our response. We agree that a proper treatment of the misspecified case requires more details to obtain the exact dependency on the misspecification parameter. The particular comment from the paper was simply meant to point out that the specific definition of $\omega(\alpha, p_0)$ is meaningful even in misspecified scenarios. We did not mean to provide formal guarantees under misspecification in the submission, and we will update the text accordingly in the final version. Does that seem reasonable to you?
> > >
> > > Thanks again for your thoughtful reading and responses!

---

> > > > ### Comment · Reviewer_ysuF · 2022-08-09
> > > > **Response on Point 1**
> > > >
> > > > Yes, I absolutely agree with the authors on that i.e a description on the misspecified case will be helpful for both our and the reader's understanding. I thank the authors for acknowledging the fact that there were no formal guarantees shown, however a discussion clearing that will help.

---

> > > > > ### Author Response · Authors · 2022-08-09
> > > > > **We will edit accordingly**
> > > > >
> > > > > Thanks again for engaging in the discussion phase! We will definitely update the discussion per your suggestion.

---

### Official Review · Reviewer_XV4J · 2022-07-14

**Rating:** 7
**Confidence:** 3
**Soundness:** 3 good
**Presentation:** 4 excellent
**Contribution:** 3 good

**Summary:**

The paper proposes a general theoretical and algorithmic framework for model-based RL in which the authors use data likelihood to estimate a model quality and use an optimistic posterior sampling for data collection.  The combination of Heliginer distance-based likelihood and posterior sampling brings some advantages for their algorithmic framework as well as for regret analysis. They show that in many special cases, the sample complexity of their proposed algorihtms improves previous related works.


**Questions:**

It would be nice if the authors address my some questions:
- What is the connection between the likelihood estimate at line 4 of the algorithm and the Hellinger distance defined in Definition 2 as the authors mentioned?
- Does the coefficient $dc(\epsilon, \alpha)$ depend on the size of the action space?
- Can the authors explain intuitively why their regret bound improves that of Sun et al. [2019] in V-type setting?
- Are your results able to compare with those using the UCB approach in model-based RL?

**Limitations:**

Yes

**Strengths And Weaknesses:**

Weaknesses:
- No experiment is provided to demonstrate the efficiency of the proposed algorithm.
- The framework is designed for only the posterior sampling approach.

Strengths:
- The paper is well written.
- The technique of combining the likelihood and optimistic posterior sampling is novel. The technique allows avoiding the minimax problem which was required in the model-free work of Agarwal and Zhang [2022], and avoiding to incorporate the Hellinger distance in the algorithm which is complicated to estimate as in the work of Foster et al. [2021].
- In some special settings, the sample complexity of their proposed general algorithm improves that of some previous works.

---

> ### Author Response · Authors · 2022-07-29
> **Thank you for the feedback. Please find responses to your questions below.**
>
> We thank you for your encouraging and constructive feedback. Indeed the main focus of our work is on a broad theoretical framework for model-based RL, and we do not conduct experiments in this work. Regarding your comment on our framework being applicable only for posterior sampling, most works on sample complexity upper bounds in RL that we are aware of analyze a specific algorithm under some assumptions. We agree that it would be great if a structural condition simultaneously enables the use of multiple algorithms, and note that several of our special cases (such as the $V$-type factorization with finite actions and the two $Q$-type factorizations) have that property. However, the theoretical analysis carried out here is indeed for our posterior sampling approach as you note.
>
> Regarding the questions:
>
> 1. By using likelihood in training, we can upper bound the squared Hellinger distance between the estimated model and the true model in the test distribution in an online fashion (see Lemma 6 in the supplementary material). We can thus use Hellinger distance to bound the Bellman error in our analysis.
>
>
> 2. Yes, the coefficient dc(\epsilon,\alpha) does depend on the size of the action space in general. In Proposition 3 for finite actions, we see a dependence on the number of actions. In Propositions 4 and 5, this dependence is not explicit, but enters through the dimension of some appropriate (state,action)-dependent features (the raw action cardinality can be infinity). In these cases, the agent can hope to generalize across similar actions without paying a raw cardinality in large and/or continuous action spaces.
>
> 3. We believe the improvement happens due to the difference in structure of our posterior sampling approach from Sun et al.'s version space technique. When Sun et al. find a policy $\pi_t$ which is at least \epsilon suboptimal, they then learn only at one level $h_t$ (like us). However, they can only guarantee that the model-based Bellman error at this step is at least $\Omega(\epsilon/H)$, which leads to a conservative setting of their parameters. We do not maintain an explicit version space and are able to set parameters in a tighter manner. The other improvement is the elimination of a $\ln |\mathcal{F}|$ factor, which does not arise in our algorithm because we do not directly estimate any integral probability metric induced by a function class $\mathcal{F}$, but instead directly perform maximum likelihood.
>
> 4. While our results are qualitatively similar to those for UCB in model-based RL, a lot of tricks which work in the specific tabular setting, such as the Bellman property of the variance used in Azar et al. (2017) are not straightforward in general non-linear settings. So we are unable to match the tightest bounds for the tabular setting, which is not surprising given the broad generality of our analysis. Beyond tabular, the algorithms of Sun et al. (2017), Du et al. (2021) can be thought of as non-linear UCB methods and our bounds improve upon them as discussed.
>
> We thank the reviewer again for their helpful comments and will update the final draft to incorporate their comments and questions with appropriate discussion.

---

### Meta-Review · Area_Chair_6r3D · 2022-08-27

**Recommendation:** Accept
**Confidence:** Certain

**Metareview:**

The paper proposes a new theoretical framework to design posterior sampling methods for model-based RL. All the reviewers agree that the theoretical framework is novel and can avoid many complications in the previous works.

**Award:**

No

---

### Decision · Program_Chairs · 2022-09-14

Accept